# Computational and cellular studies reveal structural destabilization and degradation of MLH1 variants in Lynch syndrome

Amanda B Abildgaard[1], Amelie Stein[1]*, Sofie V Nielsen[1], Katrine Schultz-Knudsen[1], Elena Papaleo[1†], Amruta Shrikhande[2], Eva R Hoffmann[2], Inge Bernstein[3], Anne-Marie Gerdes[4], Masanobu Takahashi[5], Chikashi Ishioka[5], Kresten Lindorff-Larsen[1]*, Rasmus Hartmann-Petersen[1]*

[1]Department of Biology, The Linderstrøm-Lang Centre for Protein Science, University of Copenhagen, Copenhagen, Denmark; [2]DNRF Center for Chromosome Stability, Department of Cellular and Molecular Medicine, University of Copenhagen, Copenhagen, Denmark; [3]Department of Surgical Gastroenterology, Aalborg University Hospital, Aalborg, Denmark; [4]Department of Clinical Genetics, Rigshospitalet, Copenhagen, Denmark; [5]Department of Medical Oncology, Tohoku University Hospital, Tohoku University, Sendai, Japan

*For correspondence:
amelie.stein@bio.ku.dk (AS);
lindorff@bio.ku.dk (KL-L);
rhpetersen@bio.ku.dk (RH-P)

Present address:
[†]Computational Biology Laboratory, Danish Cancer Society Research Center, Copenhagen, Denmark

Competing interests: The authors declare that no competing interests exist.

**Abstract** Defective mismatch repair leads to increased mutation rates, and germline loss-of-function variants in the repair component MLH1 cause the hereditary cancer predisposition disorder known as Lynch syndrome. Early diagnosis is important, but complicated by many variants being of unknown significance. Here we show that a majority of the disease-linked MLH1 variants we studied are present at reduced cellular levels. We show that destabilized MLH1 variants are targeted for chaperone-assisted proteasomal degradation, resulting also in degradation of co-factors PMS1 and PMS2. In silico saturation mutagenesis and computational predictions of thermodynamic stability of MLH1 missense variants revealed a correlation between structural destabilization, reduced steady-state levels and loss-of-function. Thus, we suggest that loss of stability and cellular degradation is an important mechanism underlying many *MLH1* variants in Lynch syndrome. Combined with analyses of conservation, the thermodynamic stability predictions separate disease-linked from benign *MLH1* variants, and therefore hold potential for Lynch syndrome diagnostics.
DOI: https://doi.org/10.7554/eLife.49138.001

## Introduction

The DNA mismatch repair (MMR) pathway corrects mismatched base pairs inserted during replication. The MutSα (MSH2-MSH6) heterodimer initiates repair by detecting the mismatch after which the MutLα (MLH1-PMS2) heterodimer promotes the process by generating a nick in the newly synthesized DNA strand, thereby stimulating downstream repair proteins (*Jiricny, 2006*; *Jun et al., 2006*). The MMR pathway is phylogenetically highly conserved, emphasizing its importance as a key DNA repair mechanism of the cell (*Jiricny, 2013*; *Sachadyn, 2010*). Loss of MMR activity causes genome instability, and can result in both sporadic and inherited cancer, such as Lynch syndrome (LS) (OMIM: #609310), also known as hereditary nonpolyposis colorectal cancer (HNPCC). The predominant consequence of LS is colorectal cancer (CRC), making LS the underlying reason for around 4% of all CRC cases (*Aarnio et al., 1999*; *Hampel et al., 2008*; *Sijmons and Hofstra, 2016*; *Vasen et al., 1996*; *Vasen and de Vos Tot Nederveen Cappel, 2013*; *Thompson et al., 2014*; *Møller et al., 2018*). Importantly, the cumulative lifetime cancer risk varies considerably between

patients and depends on the specific germline mutation in the genes encoding the key mismatch repair proteins MSH2, MSH6, MLH1, and PMS2 (*Barrow et al., 2008*; *Dowty et al., 2013*; *Dunlop et al., 1997*; *Lynch et al., 2015*; *Peltomäki et al., 1993*; *Plaschke et al., 2004*; *Sijmons and Hofstra, 2016*).

The majority of LS cases result from *MLH1* and *MSH2* mutations (*Peltomäki, 2016*), many of which are missense mutations (*Heinen, 2010*; *Palomaki et al., 2009*; *Peltomäki and Vasen, 1997*; *Peltomäki, 2016*). Evidently, such missense mutations may cause loss-of-function by directly perturbing protein-protein interactions or ablating enzymatic activity. Many missense mutations, however, cause loss-of-function by inducing structural destabilization of the protein (*Stein et al., 2019*), which in turn may trigger protein misfolding and degradation by the ubiquitin-proteasome system (UPS) (*Kampmeyer et al., 2017*; *Nielsen et al., 2014*; *Kriegenburg et al., 2014*). As a result, the cellular amount of a missense protein may be reduced to an insufficient level, which can ultimately cause disease (*Ahner et al., 2007*; *Casadio et al., 2011*; *Matreyek et al., 2018*; *Nielsen et al., 2017*), as we and others have previously shown for LS-linked variants of MSH2 (*Gammie et al., 2007*; *Arlow et al., 2013*; *Nielsen et al., 2017*).

In this study, we investigated whether this is the case for LS-linked variants of the MLH1 protein. We determined cellular abundance for 69 missense variants, and show that several destabilized LS-linked MLH1 variants are targeted for chaperone-assisted proteasomal degradation and are therefore present at reduced cellular amounts. In turn, this lower amount of MLH1 results in degradation of the MLH1-binding proteins PMS1 and PMS2. In silico saturation mutagenesis and computational prediction of the thermodynamic stability of all possible MLH1 single site missense variants revealed a correlation between the structural destabilization of MLH1, reduced steady-state levels and the loss-of-function phenotype. Accordingly, the thermodynamic stability predictions accurately separate disease-linked *MLH1* missense mutations from benign *MLH1* variants (area under the curve is 0.82 in a receiver-operating characteristic analysis), and therefore hold potential for classification of *MLH1* variants of unknown consequence, and hence for LS diagnostics. Further, by suggesting a mechanistic origin for many LS-causing *MLH1* missense variants our studies provide a starting point for development of novel therapies.

## Results

### In silico saturation mutagenesis and thermodynamic stability predictions

Most missense proteins are less structurally stable than the wild-type protein (*Tokuriki and Tawfik, 2009*), and individual missense variants may thus lead to increased degradation and insufficient amounts of protein. To comprehensively assess this effect for MLH1, we performed energy calculations based on crystal structures of MLH1 to predict the consequences of missense mutations in *MLH1* on the thermodynamic stability of the MLH1 protein structure. Full-length human MLH1 is a 756 residue protein which forms two folded units, an N-terminal domain (residues 7–315) and a C-terminal domain (residues 502–756) (*Mitchell et al., 2019*) separated by a flexible and intrinsically disordered linker (*Figure 1A*). Using the structures (*Wu et al., 2015*) of the two domains (PDB IDs 4P7A and 3RBN) (*Figure 1A*), we performed in silico saturation mutagenesis, introducing all possible single site amino acid substitutions into the wild-type MLH1 sequence at the 564 structurally resolved residues. We then applied the FoldX energy function (*Schymkowitz et al., 2005*) to estimate the change in thermodynamic folding stability compared to the wild-type MLH1 protein ($\Delta\Delta G$) (*Figure 1BC*). Negative values indicate mutations that are predicted to stabilize MLH1, while positive values indicate that the mutations may destabilize the protein. Thus, those variants with $\Delta\Delta G$ predictions > 0 kcal/mol are expected to have a larger population of fully or partially unfolded structures that, in turn, may be prone to protein quality control (PQC)-mediated degradation. Our saturation mutagenesis dataset comprises 19 (amino acids, excluding the wild-type residue) * 564 (residues resolved in the N- and C-terminal structures)=10,716 different MLH1 variants, thus covering 75% of all possible missense variants in MLH1. We illustrate a subsection as a heat map in *Figure 1D* (the entire dataset is included in the supplemental material, *Supplementary file 1*). The predictions reveal that 34% of the substitutions are expected to change the stability of MLH1 by less than 0.7 kcal/mol, which is the typical error of the predictions (*Guerois et al., 2002*) (*Figure 1E*). A

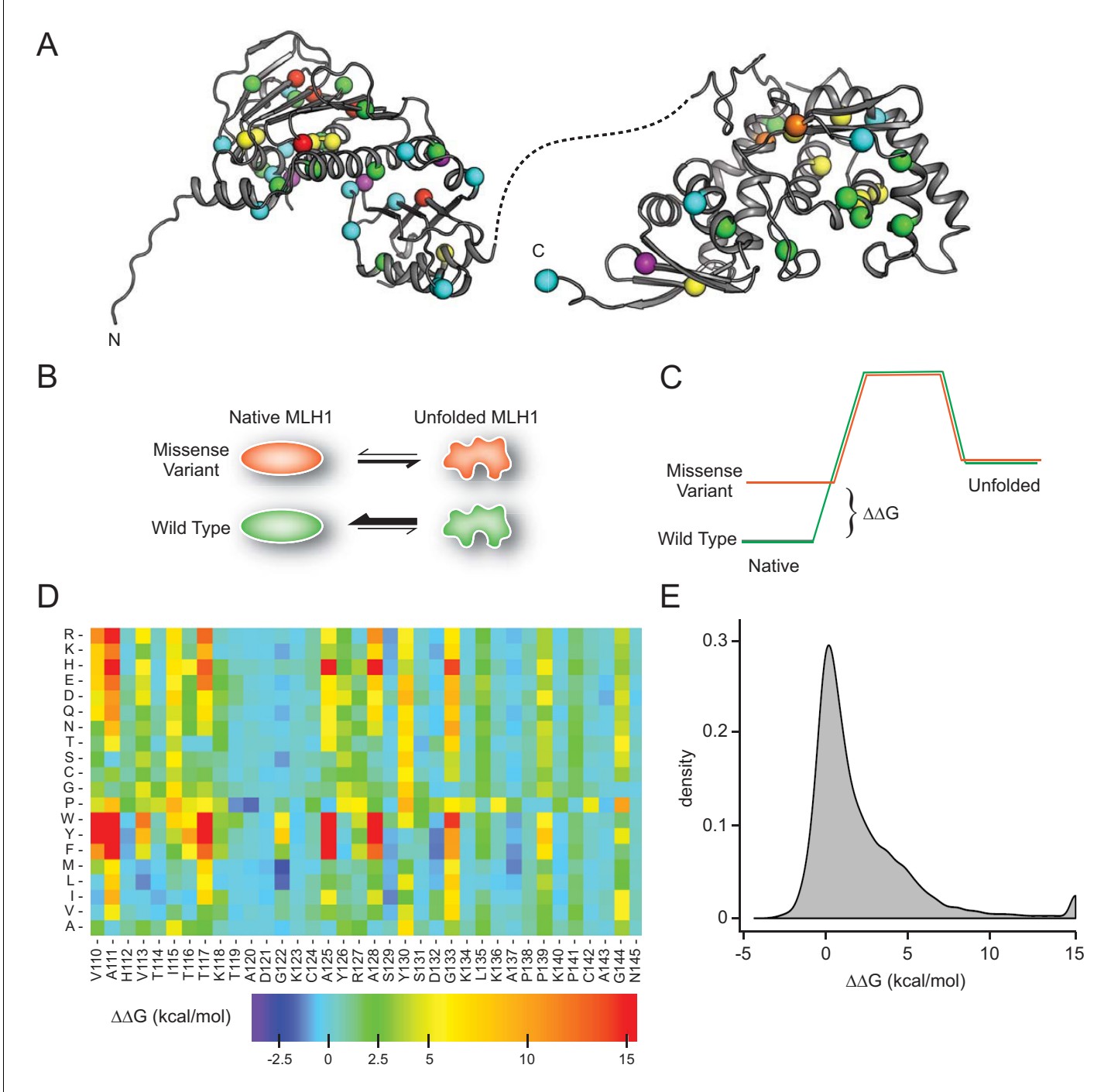

**Figure 1.** MLH1 structural stability predictions. (**A**) Structure of MLH1 (PDB IDs 4P7A and 3RBN). Positions of variants tested in this work are highlighted with colored spheres, indicating the predicted ΔΔG (<0.5 kcal/mol, purple,<1, cyan,<3.5 green,<7, yellow,<12, orange,>12, red). (**B**) Many disease-linked MLH1 missense variants (red) are structurally destabilized and therefore, compared to wild-type MLH1 (green), more likely to unfold. (**C**) The free energy of the folded conformation of a destabilized missense variant (red) is closer to that of the fully unfolded state. The employed stability calculations predict the difference of the free energy of unfolding (ΔΔG) between a missense variant (red) and wild-type MLH1 (green). (**D**) Excerpt of the in silico saturation mutagenesis map (full dataset provided in *Supplementary file 1*). (**E**) Distribution of all predicted ΔΔGs from saturation mutagenesis. The peak at 15 kcal/mol contains all variants with ΔΔG values greater than this value.

DOI: https://doi.org/10.7554/eLife.49138.002

comparable fraction (32%) are, however, predicted to cause a substantial destabilization (>2.5 kcal/mol) of the MLH1 protein (*Figure 1E*).

## Thermodynamic stability calculations predict severely reduced MLH1 steady-state levels

To test whether the in silico stability predictions are predictive of cellular stability, abundancy, and function, we selected 69 naturally occurring MLH1 missense variants with predicted ΔΔGs spanning from −1.6 to >15 kcal/mol (*Table 1*). We further ensured that the selected mutations were distributed throughout the *MLH1* gene, thus probing the entire structured parts of the MLH1 protein (*Figure 1A*). Then, the variants were introduced into *MLH1*-negative HCT116 cells and analyzed by automated immunofluorescence microscopy using a polyclonal antiserum to MLH1.

As expected, wild-type MLH1 localized primarily to the nucleus (*Figure 2A*). This localization pattern was also observed for all the MLH1 variants, and we did not detect any protein aggregates. We did, however, observe large variations in the fluorescence intensity, and consequently the steady-state protein levels, between the different MLH1 variants (*Figure 2A*). To quantify these differences, we first excluded the non-transfected cells using the intensity in the non-transfected control. Then we measured the total intensity of the MLH1 fluorescence in each cell and normalized to the intensity for wild-type MLH1. This analysis revealed up to 12-fold difference in intensity between the variants showing sizable differences in abundance.

To examine whether these variations in cellular abundance is correlated with thermodynamic stability, we plotted the normalized values against the predicted structural stabilities (ΔΔGs). This analysis indeed reveals that those MLH1 variants that were predicted to be structurally destabilized (high ΔΔGs) also displayed reduced steady-state levels (*Figure 2B*), indicating that the predicted structural destabilization and low steady-state MLH1 levels go hand in hand. Almost all (30 out of 31) variants with steady state >75% have a ΔΔG < 3 kcal/mol, and similarly most (22/23) with ΔΔG > 3 kcal/mol have steady state levels < 75%. A destabilization of 3 kcal/mol is a relatively low threshold, but consistent with previous observations of other unrelated proteins (*Nielsen et al., 2017*; *Scheller et al., 2019*; *Bullock et al., 2000*). Such low stability thresholds may indicate that local unfolding (discussed below) plays an important role in the recognition and degradation of pathogenic variants, and/or reflect that wild-type MLH1 is a marginally stable protein. To test this, cell lysates were incubated for 30 min. at a range of temperatures. Then the lysates were separated into soluble (supernatant) and insoluble (pellet) fractions by centrifugation, and the amount of soluble MLH1 was determined by blotting (*Figure 2—figure supplement 1*). Comparison with abundant cellular proteins stained by Ponceau S and blotting for GAPDH revealed that wild-type MLH1 appears somewhat less thermostable than these other proteins (*Figure 2—figure supplement 1*), supporting that MLH1 may indeed be marginally stable.

Given that decreased levels of MLH1 protein could cause loss of MMR function, we also examined whether cellular abundancy correlated with pathogenicity. Of the 69 variants that we studied, 29 are classified as pathogenic or likely pathogenic in the ClinVar database (*Landrum et al., 2018*), whereas 12 are (likely) benign, and 28 are variants of unknown significance. We found that all (likely) benign variants appeared stable and had steady-state levels > 70% (*Figure 2B*). Conversely, 18 out of the 29 pathogenic variants (62%) had steady-state levels < 70% (*Figure 2B*), suggesting that protein destabilization is a common feature for more than half of the MLH1 variants linked to LS, and that predictions of stability might be useful for classifying such variants (see below).

Next, we analyzed how the measured steady-state levels and the stability predictions correlated with previously published in vivo functional data on MLH1 (*Takahashi et al., 2007*). In that study, MLH1 function was tested in a number of assays and ranked from 0 (no function) to 3 (full function) based on their dominant mutator effect (DME) when human MLH1 variants are expressed in yeast cells (*Shimodaira et al., 1998*). Our comparison revealed that variants with reduced steady-state levels and high risk of destabilization in general are less likely to be functional (*Figure 2CD*), which again indicates that the reduced structural stability may be linked to the observed loss-of-function phenotype. For example, while 22/23 variants with DME = 3 have steady-state levels > 70%, only five of the 23 variants with DME = 0 have this high amount of protein. These functional differences are also reflected in the correlation between loss of stability (ΔΔG) and function (*Figure 2D*). In particular none of the fully functional proteins (DME = 3) are predicted to be destabilized by more than 3 kcal/mol, whereas 18/23 variants with DME = 0 are predicted to be destabilized by at least this

**Table 1.** Characteristics of the selected naturally occurring MLH1 variants.

| Variant[*] | Steady-state level (% of WT) | FoldX ΔΔG (kcal/mol) | ClinVar annotation[¤] | DME[#] |
|---|---|---|---|---|
| E23D | 90.0 | 0.49 | VUS | 3 |
| I25T | 51.3 | 2.40 | VUS | 3 |
| A29S | 109.6 | 2.06 | (likely) pathogenic | 3 |
| M35R | 72.1 | 3.52 | (likely) pathogenic | 0 |
| I36S | 63.2 | 4.08 | VUS | NA |
| N38D | 63.5 | 1.61 | VUS | 2 |
| S44F | 9.8 | >15 | (likely) pathogenic | 0 |
| S44A | 103.2 | −1.35 | VUS | 3 |
| G54E | 15.3 | >15 | VUS | 1 |
| N64S | 96.3 | 2.16 | VUS | 1 |
| G67R | 35.0 | >15 | (likely) pathogenic | 0 |
| G67W | 14.5 | >15 | (likely) pathogenic | 0 |
| I68N | 62.5 | 2.22 | (likely) pathogenic | 0 |
| R69K | 104.9 | −0.18 | VUS | 3 |
| C77Y | 61.0 | 6.57 | (likely) pathogenic | 2 |
| F80V | 72.3 | 2.22 | (likely) pathogenic | 1 |
| T82I | 100.0 | 0.54 | (likely) pathogenic | 2 |
| R100P | 46.4 | −1.25 | (likely) pathogenic | 2 |
| E102D | 97.9 | 0.34 | (likely) pathogenic | 3 |
| A111V | 68.8 | 4.96 | (likely) pathogenic | 0 |
| T117M | 32.8 | 7.14 | (likely) pathogenic | 0 |
| T117R | 46.2 | 12.70 | (likely) pathogenic | 0 |
| A128P | 62.8 | 2.40 | (likely) pathogenic | 0 |
| D132H | 110.2 | −0.30 | (likely) benign | 3 |
| A160V | 107.3 | 0.38 | VUS | 3 |
| R182G | 87.9 | 2.60 | (likely) pathogenic | 3 |
| S193P | 100.7 | 2.73 | VUS | 0 |
| V213M | 103.9 | −0.81 | (likely) benign | 3 |
| R217C | 74.1 | 1.06 | VUS | 2 |
| I219V | 112.5 | 0.66 | (likely) benign | 3 |
| I219L | 121.9 | −0.05 | (likely) benign | 3 |
| R226L | 63.3 | 0.27 | (likely) pathogenic | 1 |
| G244V | 32.0 | >15 | VUS | 0 |
| G244D | 38.8 | >15 | (likely) pathogenic | 0 |
| H264R | 117.6 | −0.60 | VUS | 3 |
| R265C | 57.2 | 0.28 | (likely) pathogenic | 2 |
| R265H | 81.4 | 0.04 | VUS | 3 |
| E268G | 81.1 | 0.81 | (likely) benign | 2 |
| L272V | 80.0 | 1.95 | VUS | 3 |
| A281V | 82.5 | 0.87 | (likely) pathogenic | 3 |
| K286Q | 101.8 | 0.28 | VUS | 2 |
| S295G | 88.6 | 0.13 | (likely) pathogenic | 2 |
| H329P | 54.1 | 5.67 | (likely) pathogenic | 1 |

*Table 1 continued on next page*

*Table 1 continued*

| Variant[*] | Steady-state level (% of WT) | FoldX ΔΔG (kcal/mol) | ClinVar annotation[¤] | DME[#] |
|---|---|---|---|---|
| V506A | 62.1 | 2.18 | VUS | 2 |
| Q542L | 110.2 | −1.56 | VUS | 3 |
| L549P | 63.7 | 5.17 | VUS | 0 |
| I565F | 65.9 | 9.64 | VUS | 0 |
| L574P | 34.4 | 11.97 | (likely) pathogenic | 0 |
| E578G | 103.2 | 0.45 | (likely) benign | 2 |
| L582V | 100.0 | 1.93 | VUS | 3 |
| L588P | 88.6 | 3.30 | VUS | 1 |
| **K618A** | 80.4 | 0.61 | VUS | 1 |
| K618T | 106.5 | 0.09 | (likely) benign | 0 |
| L622H | 61.1 | 4.97 | (likely) pathogenic | 0 |
| P640T | 61.1 | 3.78 | VUS | 0 |
| L653R | 66.1 | 3.22 | (likely) pathogenic | 0 |
| I655V | 89.1 | 1.03 | (likely) benign | 3 |
| I655T | 71.1 | 1.29 | VUS | 3 |
| **R659P** | 69.0 | 6.93 | (likely) pathogenic | 0 |
| R659Q | 84.9 | 2.41 | VUS | 2 |
| T662P | 72.7 | 5.23 | (likely) pathogenic | 0 |
| E663G | 72.0 | −0.23 | VUS | 3 |
| E663D | 97.2 | 0.66 | (likely) pathogenic | 2 |
| L676R | 36.8 | 5.12 | VUS | 0 |
| R687W | 115.4 | 1.12 | (likely) pathogenic | 0 |
| Q689R | 71.0 | −0.54 | (likely) benign | 3 |
| V716M | 100.1 | 1.41 | (likely) benign | 1 |
| H718Y | 73.7 | 0.16 | (likely) benign | 2 |
| K751R | 82.1 | −0.23 | (likely) benign | 3 |

[*]: boldfaced variants studied in detail; [¤]: VUS: variant of unknown significance; [#]DME: dominant mutator effect See also source data (***Table 1—source data 1***).

DOI: https://doi.org/10.7554/eLife.49138.003

The following source data is available for Table 1:
**Source data 1.** MLH1 variants tested in this work.
DOI: https://doi.org/10.7554/eLife.49138.004

amount. The unstable and non-functional variants do not appear structurally clustered to a particular site within the protein, and are found throughout both the N- and C-terminal domains of MLH1 (*Figure 2E*). These affected positions are, however, closer to one another than random pairs in the respective domain (average pairwise distances of 17.3 Å vs. 24.1 Å in the N-terminal domain, and 14.7 Å vs. 21.0 Å in the C-terminal domain). In contrast, the linker region is depleted in detrimental variants but not in benign variants, hence functional (*Takahashi et al., 2007*) and benign (*Landrum et al., 2018*) variants are found both in structured and unstructured regions (*Figure 2—figure supplement 2*).

## Proteasomal degradation causes reduced steady-state levels of destabilized MLH1 variants

Next, we analyzed why the steady-state levels of certain MLH1 variants were reduced. For this purpose, we carefully selected eight of the 69 missense MLH1 variants for further in-depth analyses (E23D, G67R, R100P, T117M, I219V, R265C, K618A, and R659P). As previously, these variants were

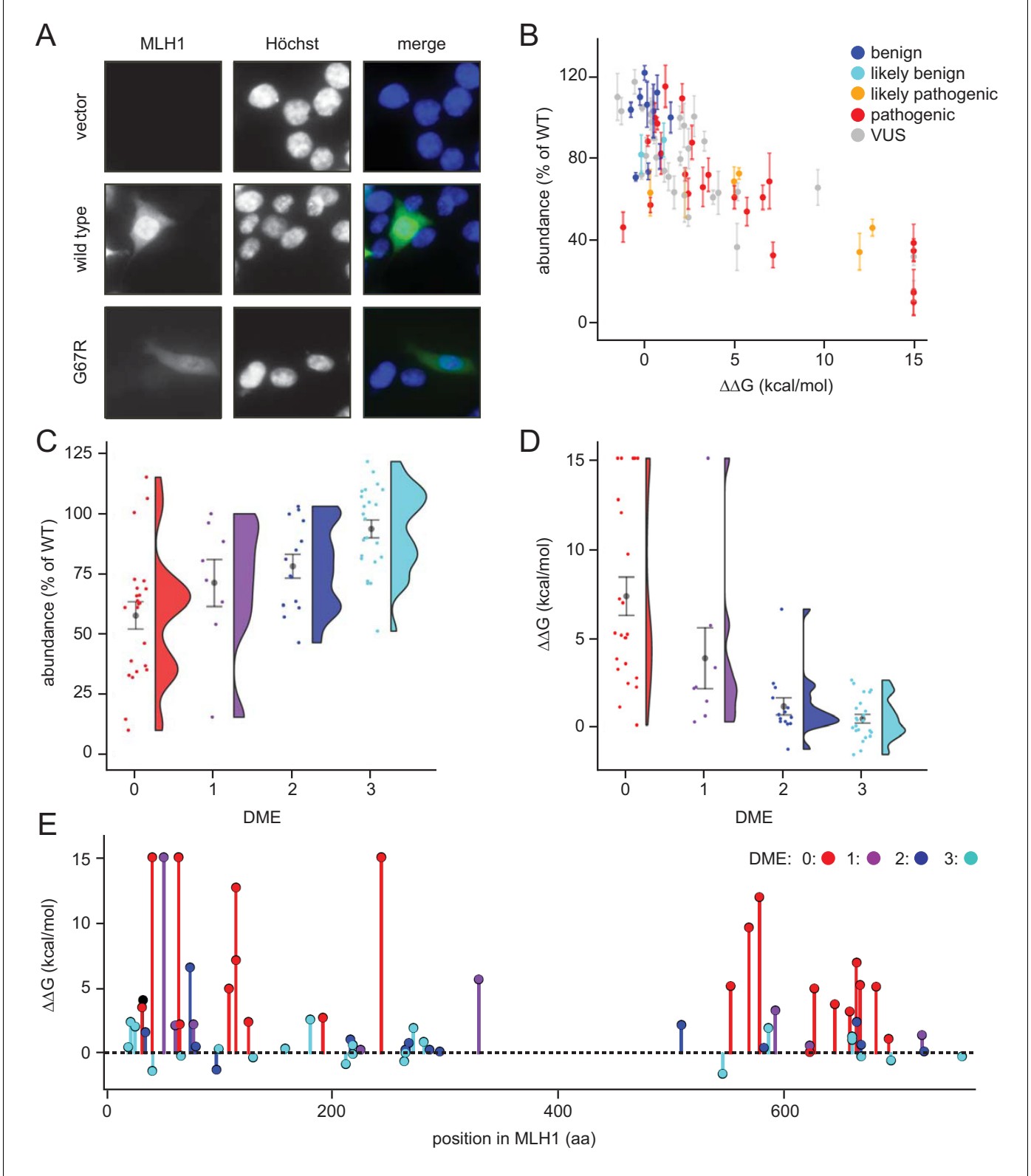

**Figure 2.** Steady-state levels of MLH1 variants correlate with structural stability predictions. (**A**) Example of the immunofluorescence imaging of HCT116 cells using antibodies to MLH1. Hoechst staining was used to mark the nucleus. Note the reduced steady-state levels of the G67R MLH1 variant compared to wild-type MLH1. (**B**) The total fluorescent intensity for each of the 69 different MLH1 variants was determined after excluding the non-transfected cells and normalizing the intensities to that for wild-type MLH1. The intensities were then plotted vs. the predicted ΔΔG values. Between

*Figure 2 continued on next page*

*Figure 2 continued*

200 and 1,000 cells were included for each quantification. The error bars indicate the standard error of the mean (n = 5 experiments). Each variant is color-coded according to the ClinVar disease category. (C) Distribution of steady-state levels by DME category – 0 is loss-of-function in all assays by *Takahashi et al. (2007)*, 3 represents function in all these assays (for details see the Materials and Methods). Raincloud plot visualization as described in *Allen et al. (2018)*. Colored surface, smoothed density estimate. Gray dots represent means within each DME category, with bars for standard error. (D) Distribution of FoldX ΔΔGs across DME categories (as in (C)). (E) FoldX ΔΔGs for all variants tested in this work, indicating their position in the MLH1 sequence. As elsewhere, values above 15 kcal/mol were here set to this value.

DOI: https://doi.org/10.7554/eLife.49138.005

The following figure supplements are available for figure 2:

**Figure supplement 1.** Solubility of wild-type MLH1 at a range of temperatures.
DOI: https://doi.org/10.7554/eLife.49138.006

**Figure supplement 2.** Variants within the central disordered region.
DOI: https://doi.org/10.7554/eLife.49138.007

chosen so the mutations were distributed across the *MLH1* gene, and to represent a broad range of predicted structural stabilities (ΔΔGs) as well as different pathogenicity annotations from the ClinVar database (*Table 1*).

The variants were transiently transfected into HCT116 cells. Indeed, six of the variants (G67R, R100P, T117M, R265C, K618A, R659P) displayed reduced steady-state levels, while wild type-like levels were observed for two variants (E23D, I219V), in agreement with the fluorescence-based observations (*Figure 3AB*). Co-transfection with a GFP-expression vector revealed that this was not caused by differences between transfection efficiencies since the amount of GFP was unchanged (*Figure 3A*).

Next, in order to investigate if the reduced MLH1 levels were caused by degradation, we monitored the amounts of MLH1 over time in cultures treated with the translation inhibitor cycloheximide (CHX). This revealed that variants with reduced steady-state levels were indeed rapidly degraded (half-life between 3 and 12 hr), whereas wild-type MLH1 and the other variants appeared stable (estimated half-life >>12 hr) (*Figure 3CD*). Treating the cells with the proteasome-inhibitor bortezomib (BZ) significantly increased the steady-state levels of the unstable variants, whereas the levels of the wild-type and stable MLH1 variants were unaffected (*Figure 3E*). Separating cell lysates into soluble and insoluble fractions by centrifugation revealed that the destabilized MLH1 variants appeared more insoluble than the stable MLH1 variants, and bortezomib treatment mainly caused an increase in the amount of insoluble MLH1 (*Figure 3—figure supplement 1*). Based on these results, we conclude that certain missense MLH1 variants are structurally destabilized, which in turn leads to proteasomal degradation and reduced steady-state protein levels, and a loss-of-function phenotype as scored by the DME.

## PMS1 and PMS2 are destabilized when MLH1 is degraded

In order to carry out its role in MMR, it is essential that MLH1 associates with PMS2 to form the active MutLα complex (*Li and Modrich, 1995*; *Räschle et al., 2002*; *Tomer et al., 2002*). Additionally, MLH1 can bind the PMS1 protein and form the MutLβ complex, the function of which remains unknown (*Cannavo et al., 2007*; *Kondo et al., 2001*; *Wu et al., 2003*).

As typical for 'orphan' proteins lacking their binding partners (*Yanagitani et al., 2017*; *McShane et al., 2016*), PMS2 has been found to be unstable in the absence of MLH1 (*Hinrichsen et al., 2017*; *Lynch et al., 2015*; *Mohd et al., 2006*; *Perera and Bapat, 2008*). To test the mechanism underlying this instability, we measured the stability of endogenous PMS1 and PMS2 in HCT116 cells with or without introducing wild-type MLH1. In cells treated with cycloheximide, the absence of MLH1 led to rapid degradation of both PMS1 and PMS2 ($t_{1/2} \sim 3$–5 hr). However, when wild-type MLH1 was present, PMS1 and PMS2 were dramatically stabilized ($t_{1/2} \sim 12$ hr) (*Figure 4AB*). Treating untransfected HCT116 cells with bortezomib led to an increase in the amount of endogenous PMS1 and PMS2, showing that their degradation is proteasome-dependent (*Figure 4C*). The stabilizing effect of MLH1 on PMS1 and PMS2 was also observed for the stable MLH1 variants (*Figure 4DE*). Accordingly, we found that the MLH1 levels correlated with the PMS1 and PMS2 levels (*Figure 4F*).

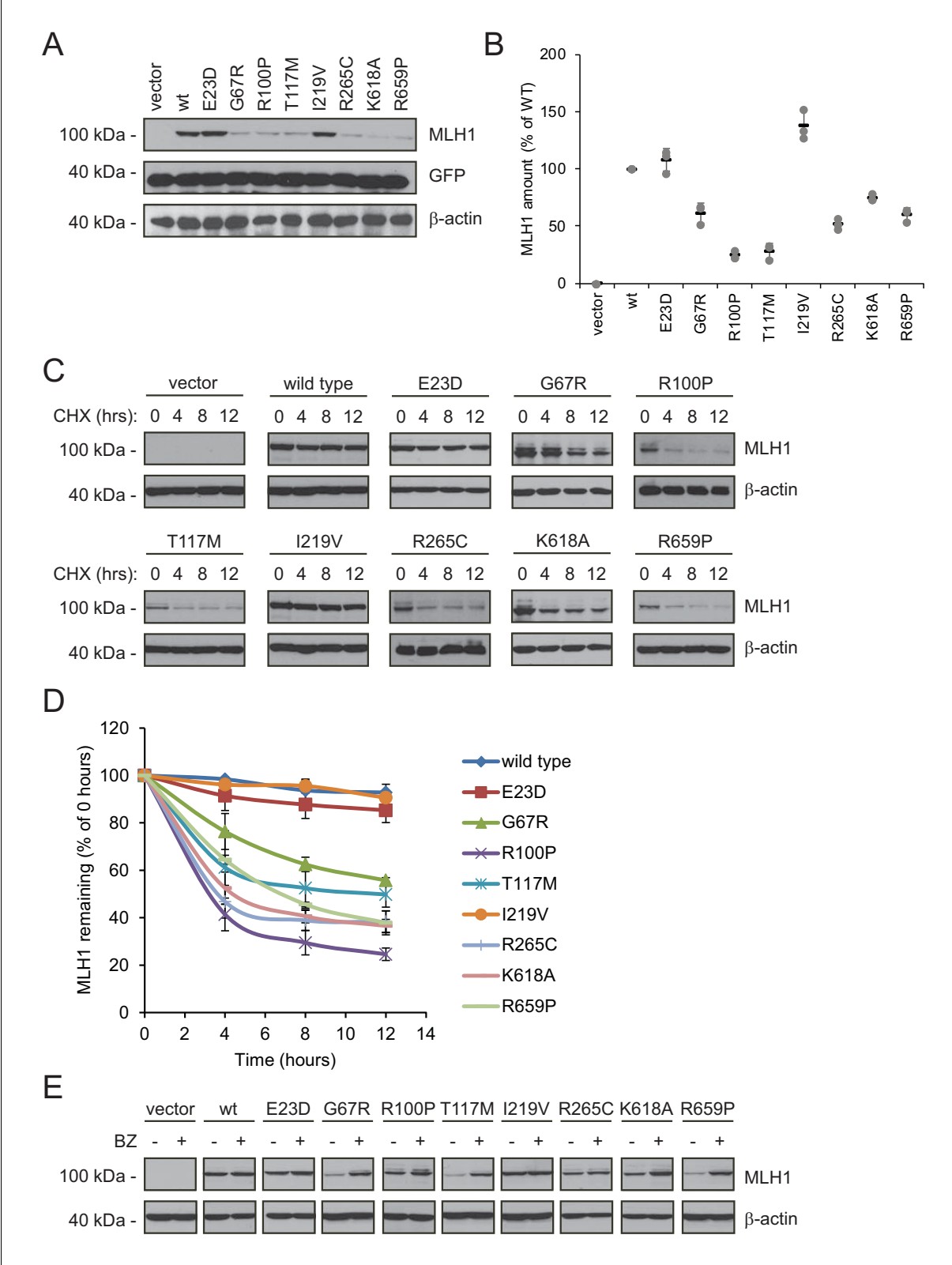

**Figure 3.** Many MLH1 variants are degraded by the proteasome. (**A**) HCT116 cells transfected with the indicated variants were analyzed by blotting with antibodies to MLH1. Co-transfection with a plasmid expressing GFP was included to test the transfection efficiencies between the MLH1 variants. β-actin served as a loading control. (**B**) Quantification of blots as in (**A**) normalized to the steady-state level of wild-type (WT) MLH1. The error bars show the standard deviation (n = 3). (**C**) MLH1-transfected HCT116 cells were treated with 25 μg/mL cycloheximide (CHX) for 0, 4, 8 or 12 hr, and lysates were

*Figure 3 continued on next page*

*Figure 3 continued*

analyzed by blotting using antibodies to MLH1. β-actin was used as a loading control. (**D**) Quantification of blots as in panel (**C**), normalized to the steady-state levels at t = 0 hr. The error bars indicate the standard deviation (n = 3). (**E**) Western blotting with antibodies to MLH1 of whole cell lysates from transfected cells either untreated or treated for 16 hr with 10 μM bortezomib (BZ). Blotting for β-actin was included as a loading control.

DOI: https://doi.org/10.7554/eLife.49138.008

The following figure supplement is available for figure 3:

**Figure supplement 1.** Solubility of selected MLH1 variants.

DOI: https://doi.org/10.7554/eLife.49138.009

Collectively, these results suggest that either there is not enough MLH1 variant in the cells to form complexes with PMS1 and PMS2 or that only stable MLH1 variants are able to bind PMS1 and PMS2, and that this binding in turn protects them from proteasomal degradation. To test these possibilities, we proceeded to assess the PMS2-binding activity of the selected MLH1 variants. To this end, HCT116 cells were co-transfected with both MLH1 and YFP-tagged PMS2. Importantly, the overexpressed YFP-PMS2 protein did not affect the MLH1 level and appeared stable in the absence of MLH1 (*Figure 4G*), allowing us to directly compare the PMS2-binding activity of the selected MLH1 variants. To ensure that the cells contained sufficient levels of the unstable MLH1 variants, the cells were treated with bortezomib prior to lysis. We found that the wild-type and stable MLH1 variants (E23D, I219V) were efficiently co-precipitated with the YFP-tagged PMS2 (*Figure 4H*). Several of the unstable MLH1 variants did not display appreciable affinity for PMS2, even after blocking their degradation, suggesting that these MLH1 variants are structurally perturbed or unfolded to an extent that disables complex formation with PMS2. Interestingly, the K618A variant displayed a strong interaction with PMS2 (*Figure 4H*), indicating that this unstable variant retains the ability to bind PMS2, and therefore potentially engage in mismatch repair. We note that this result is supported by the K618A variant's ability to stabilize PMS2 (*Figure 4E*) and the distal positioning of K618 to the PMS2 binding site (*Gueneau et al., 2013*).

## HSP70 is required for degradation of some destabilized MLH1 variants

Since structurally destabilized proteins are prone to expose hydrophobic regions that are normally buried in the native protein conformation, molecular chaperones, including the prominent HSP70 and HSP90 enzymes, often engage such proteins in an attempt to refold them or to target them for proteasomal degradation (*Arndt et al., 2007*). Indeed, both HSP70 and HSP90 are known to interact with many missense variants though with different specificities and cellular consequences (*Karras et al., 2017*), and a previous study has linked HSP90 to MLH1 function (*Fedier et al., 2005*).

To test the involvement of molecular chaperones in degradation of the selected MLH1 variants, we analyzed their interaction with HSP70 and HSP90 by co-immunoprecipitation and western blotting. Similar to above, the cells were treated with bortezomib to ensure detectable amounts of MLH1. Interestingly, four of the destabilized MLH1 variants (G67R, R100P, T117M and R265C) displayed a strong interaction with HSP70, up to approximately 7-fold greater compared to wild-type MLH1 (*Figure 5AB*). Conversely, in the case of HSP90 we observed binding to all the tested MLH1 variants, including the wild-type (*Figure 5CD*), suggesting that HSP90 may be involved in the de novo folding of wild-type MLH1 or assembly of MLH1-containing MMR complexes, while HSP70 may be involved in regulation of certain destabilized MLH1 variants, potentially playing a role in their degradation.

To test this hypothesis, we measured the steady-state levels of the MLH1 variants following inhibition of HSP70 and HSP90, respectively. We treated cells with the HSP70 inhibitor YM01 or the HSP90 inhibitor geldanamycin (GA) and compared with the MLH1 levels in untreated cells. In comparison to the HSP70 binding, the effect of YM01 appeared more subtle. The levels of three variants (G67R, R100P, T117M) were, however, increased (*Figure 5EF*), and all three were also found to bind HSP70 (*Figure 5A*) and had the lowest steady-state levels of the eight tested variants prior to HSP70 inhibition. Together, these results suggest that HSP70 actively partakes in detecting and/or directing certain destabilized MLH1 variants for degradation. We did not observe any effect of HSP90 inhibition on the MLH1 protein levels for any of the tested variants (*Figure 5GH*).

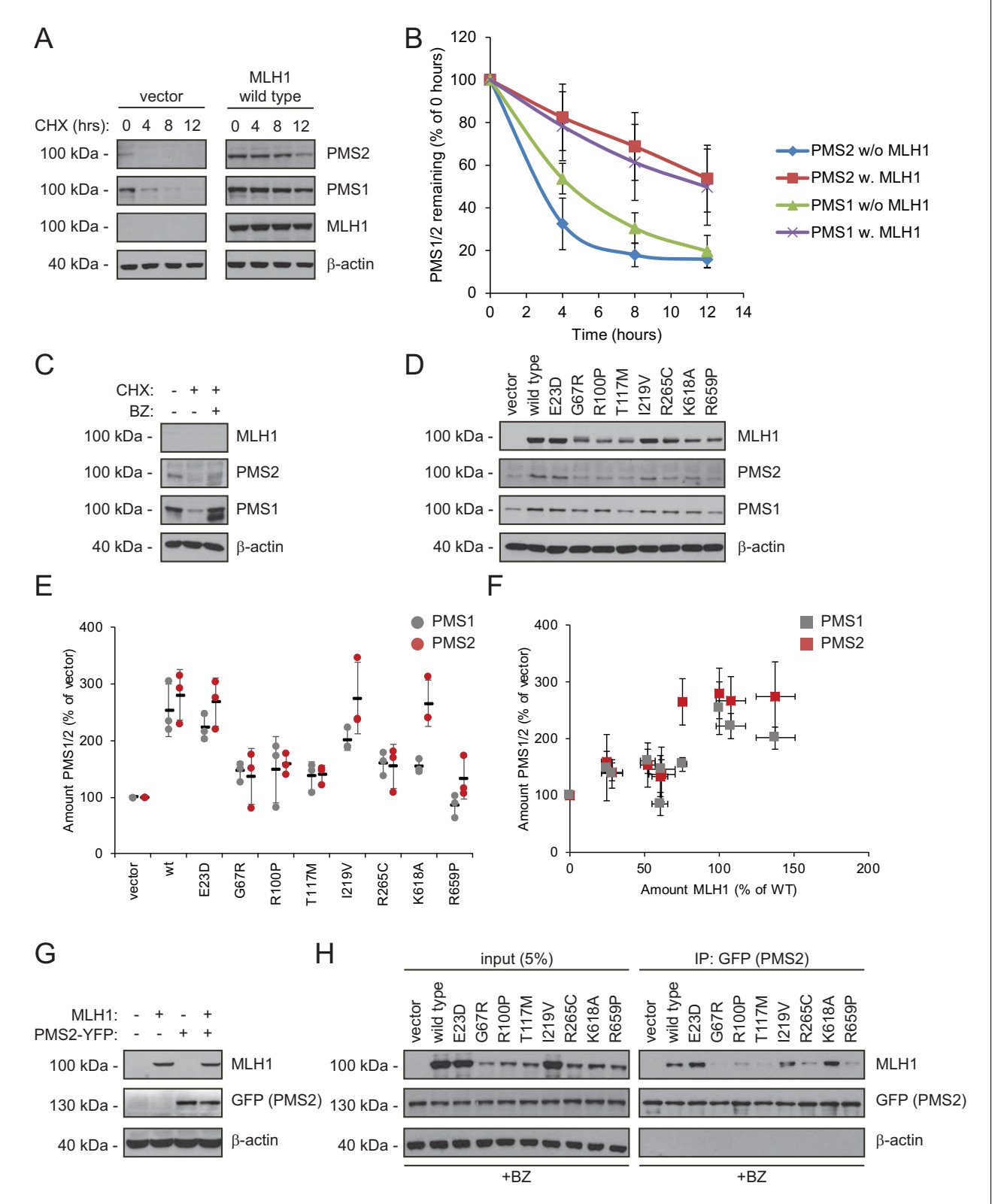

**Figure 4.** Stable MLH1 variants increase steady-state levels of PMS1 and PMS2. (**A**) The levels of endogenous PMS1 and PMS2 were determined by blotting of whole-cell lysates of HCT116 cells transfected with either empty vector or with wild-type MLH1 and treated with 25 µg/mL cycloheximide (CHX) for 0, 4, 8 or 12 hr. The antibodies used were to PMS1 and PMS2, and as a control to MLH1. β-actin served as loading control. (**B**) Quantification of blots as in panel (**A**) normalized to protein levels at 0 hr. The error bars indicate the standard deviation (n = 3). (**C**) The levels of endogenous MLH1,

*Figure 4 continued on next page*

*Figure 4 continued*

PMS1 and PMS2 were compared by blotting of cell lysates of HCT116 cells either untreated, or treated with cycloheximide (CHX) or with bortezomib (BZ) and CHX. β-actin served as loading control. (**D**) The levels of endogenous PMS1 and PMS2 and transfected MLH1 were compared by western blotting using antibodies to PMS1, PMS2 and MLH1. β-actin served as loading control. (**E**) Quantification of blots as in panel (**C**) normalized to the level of endogenous PMS1 (grey) or PMS2 (red) in untransfected HCT116 cells. The error bars show the standard deviation (n = 3). (**F**) Plotting the levels of the MLH1 variants vs. the levels of endogenous PMS1 (grey) and PMS2 (red). The error bars show the standard deviation (n = 3). (**G**) The levels of MLH1 and YFP-tagged PMS2 were analyzed by SDS-PAGE and blotting of whole-cell lysates of HCT116 cells transfected with the indicated expression plasmids. β-actin was included as loading control. (**H**) Co-transfected PMS2-YFP was immunoprecipitated (IP) using GFP-trap beads, and the precipitated material was analyzed by electrophoresis and blotting. Bortezomib was added to all cultures 16 hr prior to cell lysis to ensure ample amounts of the unstable MLH1 variants.

DOI: https://doi.org/10.7554/eLife.49138.010

## Structural stability calculations for predicting pathogenic mutations

Our results show that unstable protein variants are likely to be rapidly degraded, suggesting that predictions of changed thermodynamic stability of missense MLH1 variants could be used to estimate whether a particular MLH1 missense variant is pathogenic or not. In comparison with the sequence-based tools (e.g. PolyPhen2, PROVEAN and REVEL) that are currently employed in the clinic (*Adzhubei et al., 2010*; *Choi and Chan, 2015*), the FoldX energy predictions provide an orthogonal structure-based and sequence-conservation-independent prediction of whether a mutation is likely to be pathogenic. Unlike most variant consequence predictors, FoldX was not trained on whether mutations were benign or pathogenic, but solely on biophysical stability measurements (*Guerois et al., 2002*). This considerably reduces the risk of overfitting to known pathogenic variants. More importantly, because of the mechanistic link to protein stability, FoldX predictions enable insights into why a particular mutation is problematic (*Kiel and Serrano, 2014*; *Kiel et al., 2016*; *Pey et al., 2007*; *Nielsen et al., 2017*; *Stein et al., 2019*).

As a first test for utilizing the biophysical calculations, we analyzed the predicted protein stabilities of MLH1 variants reported in the >140.000 exomes available in the Genome Aggregation Database (gnomAD) (*Lek et al., 2016*; *Karczewski et al., 2019*). Gratifyingly, this revealed that those variants reported to occur at a high frequency in the population all displayed low ΔΔG values (*Figure 6A*), suggesting that these MLH1 proteins are stable. Accordingly, with only a few exceptions, the most common MLH1 alleles reported in gnomAD also appeared functional (high DME scores) (*Figure 6A*). To further test the performance of the structural stability calculations for identifying pathogenic MLH1 variants, we then compared the ΔΔG values for ClinVar-annotated MLH1 variants. This revealed that the benign MLH1 variants all appeared structurally stable, while many pathogenic variants appeared destabilized (*Figure 6B*). For example, in our dataset 15 of the 28 pathogenic variants (54%) for which we could calculate a stability change have ΔΔG > 3 kcal/mol and 20 (71%) have ΔΔG > 2 kcal/mol, whereas none of the 11 benign variants have ΔΔG > 1.5 kcal/mol. Applying this calculation to all MLH1 missense variants in ClinVar revealed similar trends, with 55/95 (57%) of the pathogenic variants having ΔΔG > 3 kcal/mol, while only 2/21 benign variants have ΔΔG > 1.5 kcal/mol.

Overall, while many pathogenic variants are severely destabilized, a subset of these are predicted to be as stable as non-pathogenic variants (e.g. 7/28 have predicted ΔΔG < 1.0 kcal/mol). This observation could be explained for example by inaccuracies of our stability calculations or by loss of function via other mechanisms such as direct loss of enzymatic activity, post-translational modifications or protein-protein interactions (*Wagih et al., 2018*). Thus, as a separate method for predicting the biological consequences of mutations, we explored if sequence analysis of the MLH1 protein family across evolution would reveal differences in selective pressure between benign and pathogenic variants. We performed an analysis of a multiple sequence alignment of MLH1 homologs, which considers both conservation at individual sites, but also non-trivial co-evolution between pairs of residues (*Balakrishnan et al., 2011*; *Stein et al., 2019*). Turning this data into a statistical model allowed us to score all possible missense MLH1 variants. As this statistical sequence model is based on homologous sequences shaped by evolutionary pressures, it is expected to capture which residues, and pairs of residues, are tolerated (*Balakrishnan et al., 2011*). As opposed to stability calculations via for example FoldX, this approach is not directly linked to an underlying mechanistic model. Thus, we generally expect destabilizing residues to be recognized as detrimental by both

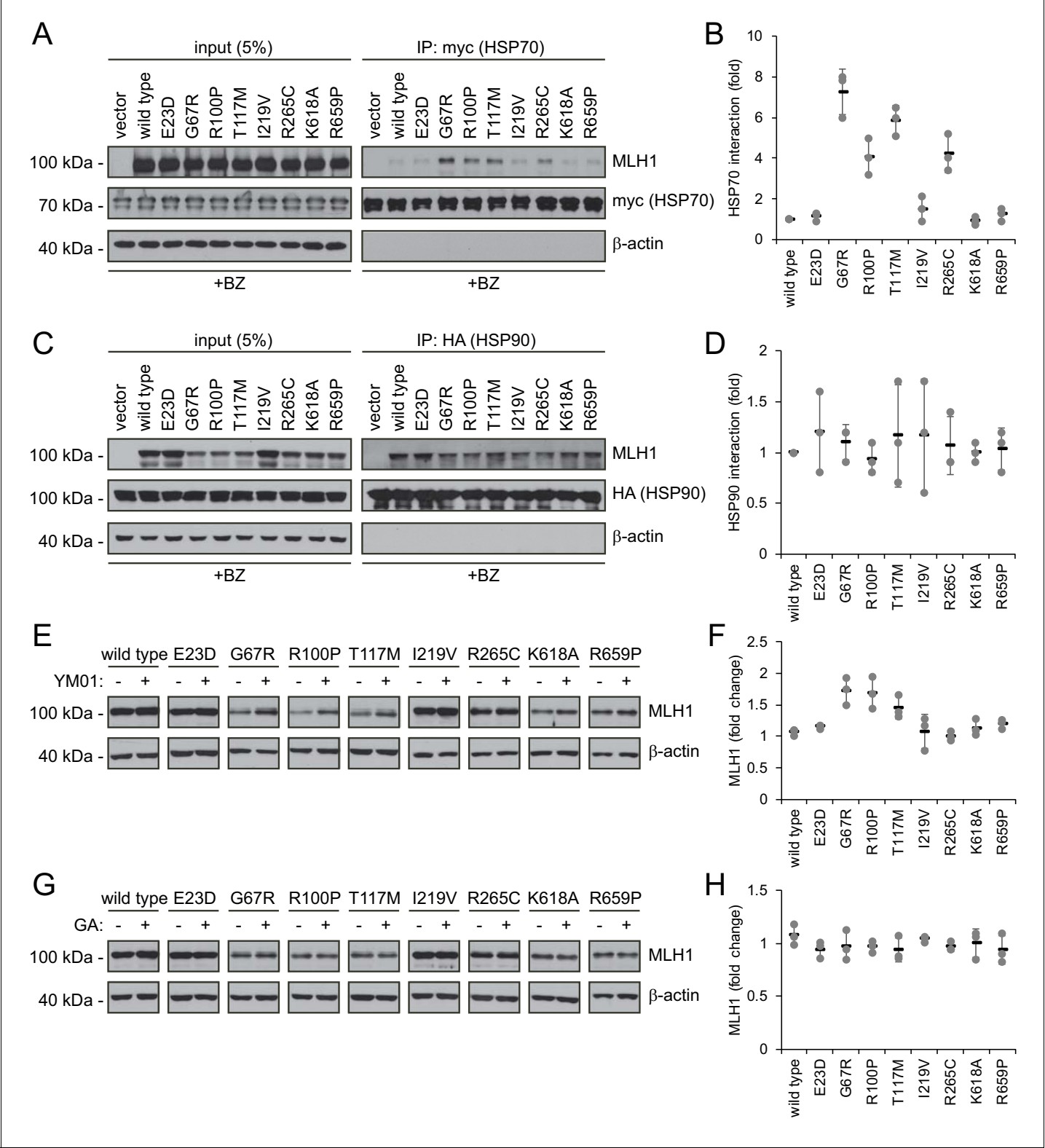

**Figure 5.** Molecular chaperones play a role in the proteasomal degradation of MLH1. (**A**) Co-transfected HSP70-myc was immunoprecipitated (IP) using myc-trap beads and analyzed by blotting with antibodies to the myc-tag (HSP70) and MLH1. Bortezomib was added to all cultures 8 hr prior to cell lysis to ensure ample amounts of the unstable MLH1 variants. (**B**) Quantification of blots as shown in panel (**A**) normalized to level of wild-type MLH1. The error bars indicate the standard deviation (n = 3). (**C**) Co-transfected HSP90-HA was immunoprecipitated (IP) with anti-HA resin, and the precipitated

*Figure 5 continued on next page*

*Figure 5 continued*

material analyzed by electrophoresis and western blotting using antibodies to the HA-tag (HSP90) and MLH1. As above, bortezomib was added to all cultures prior to cell lysis. (**D**) Quantification of blots as in panel (**C**) normalized to amount of precipitated wild-type MLH1. The error bars show the standard deviation (n = 3). (**E**) Western blotting using antibodies to MLH1 of whole-cell lysates from transfected cells treated with 5 µM YM01 for 24 hr as indicated. (**F**) Quantification of blots as shown in panel (**E**) normalized to level of MLH1 without YM01. The error bars indicate the standard deviation (n = 3). (**G**) Western blotting using antibodies to MLH1 of whole-cell lysates from transfected cells treated 1 µM geldanamycin (GA) for 24 hr. (**H**) Quantification of blots as shown in panel (**G**) normalized to level of MLH1 without GA. The error bars indicate the standard deviation (n = 3).

DOI: https://doi.org/10.7554/eLife.49138.011

FoldX and the evolutionary sequence energies, while variants in functionally active sites might only be recognized by the latter, if they do not affect protein stability (*Stein et al., 2019*). On the other hand, stability calculations could capture effects specific to human MLH1 that are more difficult to disentangle through the sequence analyses. In our implementation, low scores indicate mutations that during evolution appear tolerated, while high scores mark amino acid substitutions that are rare and therefore more likely to be detrimental to protein structure and/or function. Indeed, the average sequence-based score for the benign variants is lower (variations more likely to be tolerated) than the average for the ClinVar-curated pathogenic variants (*Figure 6C*). The full matrix of evolutionary sequence energies is included in the supplemental material (*Supplementary file 2*).

Next, we compared the structure-based stability calculations and evolutionary sequence energies in a two-dimensional landscape of variant tolerance (*Figure 6D*), which we find largely agrees with the functional classification by *Takahashi et al. (2007)*. There are, however, three variants with low evolutionary sequence energies (typically indicating tolerance), but predicted and experimentally-confirmed to be destabilized relative to wild-type MLH1 (T662P, I565F, G244V). Further, a number of stable variants have high DME scores (indicating wild-type-like function), but also high evolutionary sequence energies, indicating likely loss of function, and indeed several of these are classified as pathogenic in ClinVar (*Figure 6—figure supplements 1* and *2*). One possible explanation for these discrepancies is a different sensitivity in the employed yeast assays (*Takahashi et al., 2007*), that is these variants may be sufficiently functional under assay conditions, but their impaired function relative to wild-type MLH1 may nevertheless render them pathogenic in human variant carriers.

To exploit the complementary nature that ΔΔG and evolutionary sequence energies display for a subset of the variants, we applied logistic regression to combine these two metrics. In a jackknife test (to avoid overfitting) we found that 99 of 116 (85%) ClinVar (*Landrum et al., 2018*) missense variants were classified correctly by the regression model (*Figure 6E*).

To compare the capability of the above-described evolutionary sequence energies, FoldX, our regression model, and more traditional sequence-based methods (PolyPhen2, PROVEAN and REVEL) in separating pathogenic and non-pathogenic variants, we applied these approaches to the set of 116 known benign and disease-causing MLH1 variants. We then used receiver-operating characteristic (ROC) analyses to compare how well the different methods are able to distinguish the 21 benign variants from the 95 known pathogenic variants (*Figure 6F* and *Figure 6—figure supplement 3*). The results show that although all predictors perform fairly well, the logistic regression model performs best (AUC: 0.90 ± 0.03), and evolutionary sequence energies alone (AUC: 0.88 ± 0.03) are slightly better at distinguishing disease-linked missense variants from harmless variants than REVEL (AUC: 0.83 ± 0.03), PolyPhen2 (AUC: 0.84 ± 0.03) and PROVEAN (AUC: 0.76 ± 0.04). Structure-based ΔΔG calculations show similar performance to these sequence-based predictors (AUC: 0.82 ± 0.04), but, as the shape of the ROC curve illustrates, are particularly informative in the region of high specificity.

Lastly, we assessed whether the underlying genomic changes could affect splicing and thus have pathogenic potential, rather than directly acting on the protein level. Using SpliceAI (*Jaganathan et al., 2019*) we predicted that 19/116 (16%) of the ClinVar variants may affect splicing. All 19 are pathogenic variants; no benign variants were predicted to affect splicing. Interestingly, 4 of these variants have scores in our regression model that suggest them to be non-detrimental, and thus predicted not to affect protein function or stability (*Figure 6G*). Hence, more than half (4/7) of these erroneously classified variants may affect splicing rather that protein function, a substantially higher fraction than the 17% (15/88) that are predicted to affect splicing among those variants that are correctly classified by our regression model integrating ΔΔG and evolutionary sequence

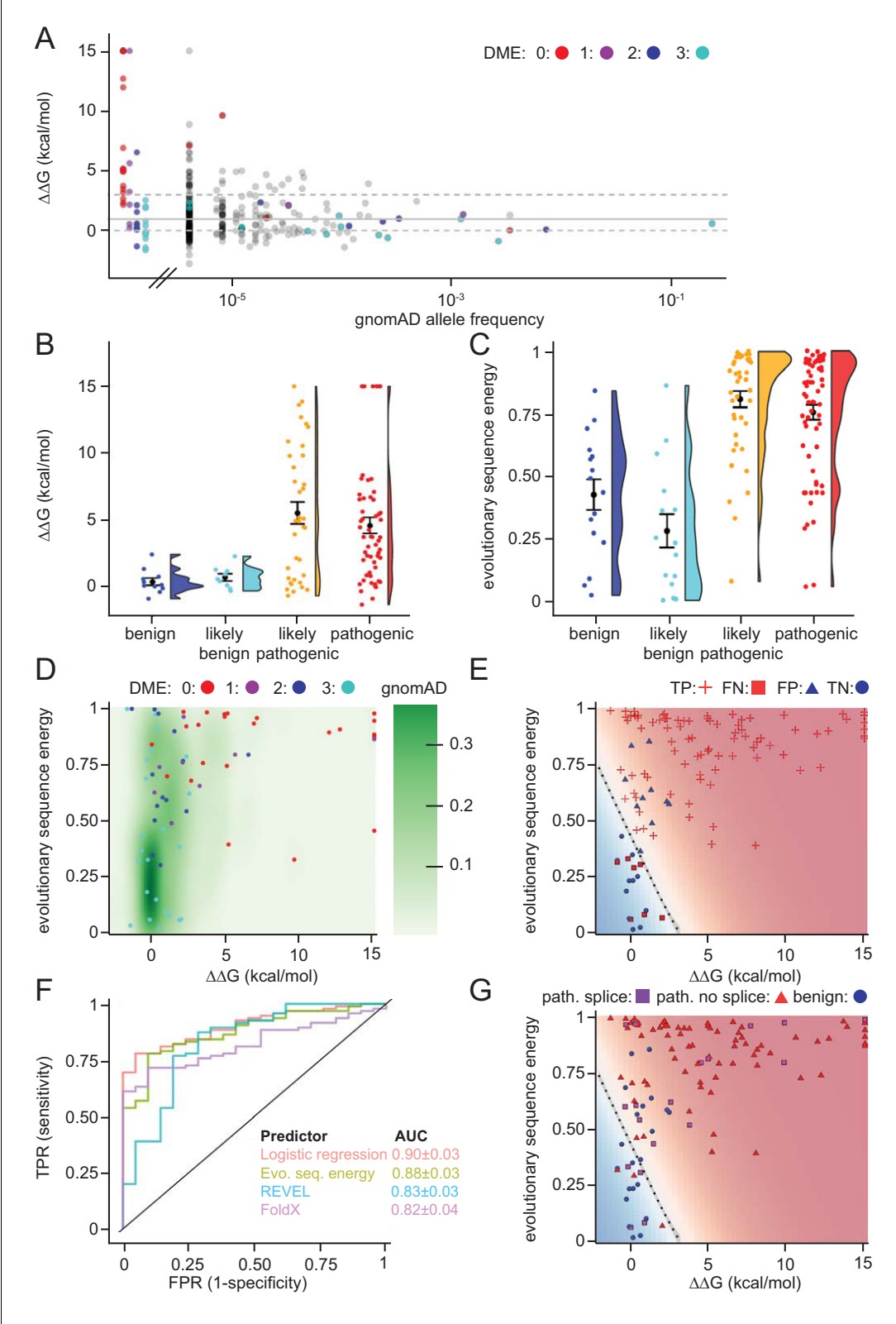

**Figure 6.** Assessing stability calculations for predicting pathogenicity. (**A**) 'Fishtail plot' of ΔΔG-values vs. allele frequencies for all variants listed in gnomAD (gray), as well as those analyzed by *Takahashi et al. (2007)*; the latter are color-coded by DME. Note that the leftmost group of colored dots are variants that have been reported in patients, but are not recorded in gnomAD (thus their allele frequency in gnomAD is zero). Variants with common to intermediate frequencies are all predicted to be stable, while some rare variants are predicted to be destabilized. ΔΔGs for gnomAD

*Figure 6 continued on next page*

*Figure 6 continued*

variants are provided as source data (*Figure 6—source data 1*), those for variants characterized by *Takahashi et al. (2007)* in *Table 1* and source data (*Table 1—source data 1*). (B) FoldX ΔΔG for benign (blue), likely benign (cyan), likely pathogenic (orange), and pathogenic (red) variants that are reported in ClinVar with 'at least one star' curation. The whiskers represent the mean and standard error of the mean. (C) Evolutionary sequence energies for ClinVar-reported variants, color scheme as in (B). The whiskers represent the mean and standard error of the mean. (D) Landscape of variant tolerance by combination of changes in protein stability (x axis) and evolutionary sequence energies (y axis), such that the upper right corner indicates most likely detrimental variants, while those in the lower left corner are predicted stable and observed in MLH1 homologs. The green background density illustrates the distribution of all variants listed in gnomAD. The combination of metrics captures most non-functional variants (DME scores 0 or 1). Outliers are discussed in the main text. (E) Logistic regression model of FoldX ΔΔGs and evolutionary sequence energies. Pathogenic variants in red, benign in blue. Dot shape indicates whether pathogenicity of the respective variant was correctly predicted by a regression model trained on all but this data point ('jackknife', TP, true positives, FN, false negatives, FP, false positives, TN, true negatives). Parameters for a model trained on the full dataset are: FoldX ΔΔG weight 0.52, evolutionary sequence energy weight 3.50, intercept −1.55. (F) ROC curves for logistic regression model, FoldX ΔΔGs, evolutionary sequence energies, and the ensemble-predictor REVEL to assess their performance in separating benign from pathogenic variants. TPR, true positive rate. FPR, false positive rate. Standard deviations in AUC were determined by performing 100 ROC analyses on randomly sampled but balanced subsets, so that there are equal numbers of positive and negative cases. (G) Integrating potential effects these variants may have on splicing in the genomic context. Purple squares indicate pathogenic variants that are predicted to affect splicing (SpliceAI, threshold 0.5). No benign variants are predicted to affect splicing. Mapping to genomic loci (*Yates et al., 2015*) and SpliceAI Scores for ClinVar entries used in this work are provided as source data (*Figure 6—source data 2*).

DOI: https://doi.org/10.7554/eLife.49138.012

The following source data and figure supplements are available for figure 6:

**Source data 1.** Data for gnomAD variants.
DOI: https://doi.org/10.7554/eLife.49138.016
**Source data 2.** Data for ClinVar variants.
DOI: https://doi.org/10.7554/eLife.49138.017
**Figure supplement 1.** Landscape of ClinVar MLH1 variant tolerance.
DOI: https://doi.org/10.7554/eLife.49138.013
**Figure supplement 2.** Positioning of selected variants near the active site in the N-terminal domain.
DOI: https://doi.org/10.7554/eLife.49138.014
**Figure supplement 3.** ROC curves for variant consequence predictors tested in this work.
DOI: https://doi.org/10.7554/eLife.49138.015

energies. The overall fraction predicted to affect splicing is similar to that reported in a recent genome-editing-based study on BRCA1 variants (*Findlay et al., 2018*).

## Discussion

Missense variants in the *MLH1* gene are a leading cause of Lynch syndrome (LS) and colorectal cancer (*Peltomäki, 2016*). In recent years, germline mutations that cause structural destabilization and subsequent protein misfolding have surfaced as the cause of several diseases, including cystic fibrosis (*Ahner et al., 2007*), phenylketonuria (*Pey et al., 2007*; *Scheller et al., 2019*), early onset Parkinson's disease (*Mathiassen et al., 2015*; *Olzmann et al., 2004*) and MSH2-linked LS (*Arlow et al., 2013*; *Nielsen et al., 2017*). Although previous studies have shown some MLH1 variants to be destabilized (*Takahashi et al., 2007*; *Perera and Bapat, 2008*), this has not been systematically addressed and the contribution of MLH1 protein stability for LS remains to be resolved.

In this study, we performed in silico saturation mutagenesis and stability predictions of all single-site MLH1 missense variants in the structurally-resolved regions of MLH1. Comparisons with a selected group of naturally occurring MLH1 variants revealed that those variants that are predicted to be destabilized indeed display substantially reduced steady-state protein levels. The decreased cellular amounts are caused by rapid proteasomal degradation. In turn, the loss of MLH1 causes a dramatic destabilization and proteasomal degradation of both PMS1 and PMS2. This effect suggests that the MutLα and MutLβ heterodimers are likely to be rather stable protein complexes, as the PMS1 and PMS2 proteins would otherwise be required to be stable in the absence of MLH1. These observations are in line with several previous studies on individual MLH1 variants (*Cravo et al., 2002*; *Kosinski et al., 2010*; *Perera and Bapat, 2008*; *Raevaara et al., 2005*) including a thorough analysis by *Takahashi et al. (2007)*, and also agree with tissue staining of tumor cells from patients with germline *MLH1* mutations (*Hampel et al., 2008*; *de Jong et al., 2004*).

Our observation that single amino acid changes in MLH1 are sufficient to cause degradation is similar to results from multiple other proteins including recent deep mutational scans on PTEN and TPMT (*Matreyek et al., 2018*), our previous results on MSH2 in human cells (*Nielsen et al., 2017*), and earlier observations on Lynch syndrome MSH2 variants in yeast (*Gammie et al., 2007*; *Arlow et al., 2013*). Our structural stability calculations predict that a relatively mild destabilization of just a few (~3) kcal/mol is sufficient to trigger MLH1 degradation, an observation in line with previous studies on other proteins (*Bullock et al., 2000*; *Nielsen et al., 2017*; *DDD Study et al., 2017*; *Scheller et al., 2019*; *Jepsen et al., 2019*; *Caswell et al., 2019*). Although the absolute thermodynamic stability of MLH1 is unknown, both in vitro and in a cellular context, it is possible that the 3 kcal/mol destabilization necessary to trigger degradation is lower than that required to reach the fully unfolded state. Although we show that wild-type MLH1 is perhaps only marginally stable, we propose that these effects of slightly destabilizing amino acid substitutions could be the result of local unfolding events rather than global unfolding. Accordingly, it appears that the PQC system is tightly tuned to detect increased amounts of minor or transient structural defects. This is supported by observations on MSH2 showing that in some cases the destabilized variants are even functional when degradation is blocked (*Arlow et al., 2013*), and results showing that the PQC system preferably targets folding intermediates (*Bershtein et al., 2013*). As a result, it is surprising that predictions of changes to the global folding stability (using the fully unfolded state as a reference) are so effective in distinguishing variants that are stable in the cell from those that are more rapidly degraded. These observations also suggest that improved understanding of the biophysical basis for cellular quality control might lead to even better predictions of cellular abundance.

In our cellular studies, the MLH1 variants were expressed from a constitutive promoter, thus bypassing any potential transcriptional regulation of MLH1 expression involving, for example, amounts of the MLH1 protein or its function. Hence, in other experimental setups where MLH1 variants are generated at the endogenous locus, additional layers of control may affect the observed correlation between structural stability and steady-state amounts. Another main difference between the cellular context and the predicted stability is the multiple roles played by the PQC system. For example, on one hand, chaperones may aid in folding or stabilizing proteins, but may at the same time act as sensors of misfolded proteins and help target them for degradation. Increased amounts of unfolded or misfolded proteins may affect and titrate PQC components, thus complicating the relationship between protein stability and abundance. Nevertheless, since the predicted protein stabilities can, to a large extent, discriminate between pathogenic and benign variants, we expect that multiple disease-linked MLH1 variants will indeed display reduced cellular levels.

Similar to our previous observations for MSH2 (*Nielsen et al., 2017*), these results indicate that structural destabilization appears to be a common consequence of many disease-linked *MLH1* missense variants. Supported by earlier functional studies (*Takahashi et al., 2007*), we suggest that the loss-of-function phenotype in many cases can be explained by structural destabilization and subsequent degradation. Indeed, 20 out of the 31 variants with DME = 0 or DME = 1 have steady-state levels less than 70% of wildtype MLH1. Our data also include examples of loss-of-function variants with high steady-state levels, which is expected, as variants can affect function without modifying stability, for example, by changing binding interfaces or active sites (*Gueneau et al., 2013*), and these would therefore be interesting to analyze biochemically in more detail. Hence, M35R, N64S and F80V are all close to the ATP binding site in the N-terminal domain of MLH1 (*Figure 6—figure supplement 2*), and might thus interfere with the catalytic activity. This would be consistent with a loss-of-function phenotype (*Takahashi et al., 2007*) but wild-type-like cellular protein levels (*Table 1*).

The correlation between the predicted structural stability with both cellular stability and protein function suggests that the stability predictions may be used for classifying MLH1 missense variants. This is particularly relevant for LS, where according to the ClinVar database (*Landrum et al., 2018*) 711 out of 851 (~84%) reported MLH1 missense variants are assigned as so-called variants of uncertain significance (VUS) (*Manolio et al., 2017*). Of note, the fact that we did not observe the E23D VUS as destabilized does not preclude this variant from being pathogenic, since it may affect function without being structurally perturbed. Although the variant appears functional in yeast cells (*Takahashi et al., 2007*), the evolutionary sequence energy of 0.9 indicates that this change is rare across the MLH1 protein family evolution and thus might be detrimental. Our results for the K618A VUS suggest that this variant, albeit being unstable, is still able to associate with PMS2 and may

therefore be functional, but untimely degraded. Of the 69 MLH1 variants that we analyzed, 30 have status as VUSs. Of these, our analysis identified several (e.g. G54E, G244V, and L676R) that hold characteristics indicating that they are likely pathogenic: steady-state levels below 50% of WT, high $\Delta\Delta G$ and low functionality score in vivo (*Takahashi et al., 2007*) (*Table 1*).

The potential use of stability predictions for LS diagnostics is supported by the predicted MLH1 stabilities clearly separating into disease-linked and benign MLH1 variants. Moreover, since we observe that those MLH1 alleles that occur more frequently in the population are in general predicted as stable, this suggests that these common MLH1 alleles are either benign or at least only disease-causing with a low penetrance. However, certainly not all the unstable variants were accurately detected by the structural predictions. For instance, out of our eight selected variants, three (R100P, R265C, K618A) appeared unstable, but were not predicted to be so (*Table 1*). As described above, it is important to note that the stability predictions report on the global stability of the protein, while in a cellular context it is unlikely that any of the variants are fully unfolded. Instead, it is likely that local elements unfold (*Stein et al., 2019*), and although refolding may occur, the locally unfolded state allows chaperones and other protein quality control components to associate and target the protein for proteasomal degradation (*Figure 7*). A better understanding of the importance of local unfolding events for cellular stability is an important area for further research. Indeed, in addition to the utility for classifying potential LS variants, we believe that an important aspect of our work is that it suggests a single mechanistic origin of ~60% of the LS MLH1 variants that we have studied. This observation is also supported by our ROC analysis (*Figure 6D*), which demonstrates that the stability calculations can identify ~60% of the pathogenic variants at very high specificity (few false positives)

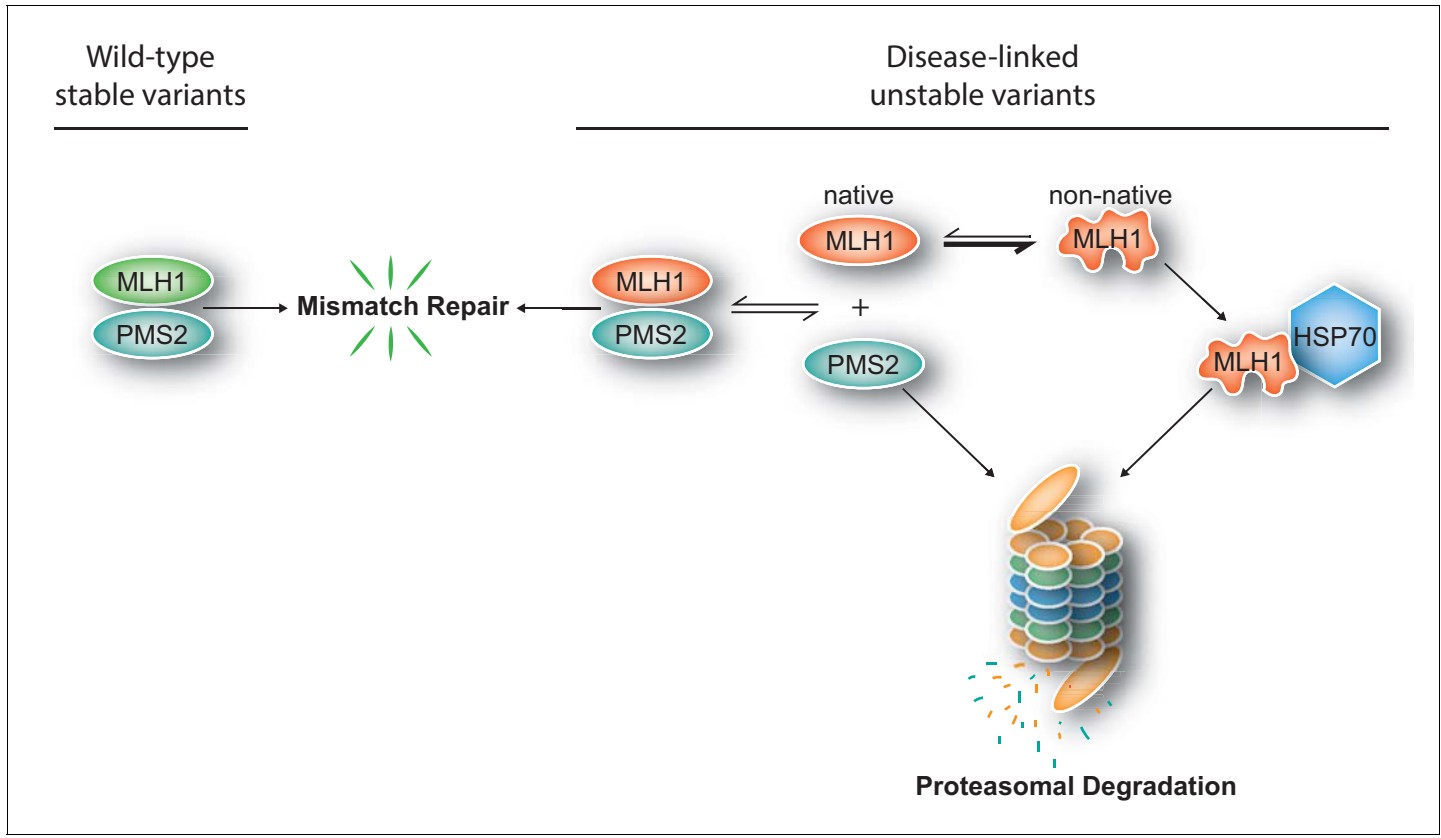

**Figure 7.** Model for how structural destabilization of MLH1 contributes to disease. The wild-type (green) MLH1-PMS2 heterodimer promotes DNA mismatch repair. Disease-linked missense MLH1 variants (red) may also promote DNA repair, but are at risk of dissociating from PMS2 due to structural destabilization. The structural destabilization of MLH1 may also cause a partial unfolding of MLH1 which is recognized by the molecular chaperone HSP70 and causes proteasomal degradation of the MLH1 variant. In turn, the degradation of MLH1 leaves PMS2 without a partner protein, resulting in proteasomal degradation of PMS2.
DOI: https://doi.org/10.7554/eLife.49138.018

exactly because these are the variants that appear to cause disease via this mechanism. The ability to separately analyze effects on protein stability and other effects that might be captured by the sequence analysis is also an advantage of applying the predictors individually, rather than relying on combined predictors such as our regression model or published ensemble-based meta-predictors. While those have slightly higher overall accuracy, they do not directly indicate the underlying molecular reason for pathogenicity. Incidentally, we note that by analyzing both calculations and multiplexed assays of variant effects, we recently found that ~60% of disease-causing variants in the protein PTEN were caused by destabilization and a resulting drop in cellular abundance (*Jepsen et al., 2019*). On the other hand, while stability prediction is very useful for accurate identification of many pathogenic variants, it may have lower overall sensitivity. We speculate that this is due to stability being a necessary, but not sufficient criterion. Thus, destabilized variants are likely pathogenic, while a variant being stable does not necessary imply it being functional.

Lastly, we predicted the effects of variants with known consequences (benign or pathogenic) may have on splicing, which may also lead to pathogenic changes. We found that, overall, about 16% of the variants are predicted to affect splicing, though none of the 19 benign variants are in this category. Interestingly, 4/7 pathogenic variants in the region that the logistic regression model predicts to be benign may affect splicing (*Figure 6G*), indicating that integration of effects on the genomic level is likely to boost overall predictive power, and should be considered in future developments of pathogenicity predictors.

In line with observations on other destabilized proteins, we found that the degradation of some structurally destabilized MLH1 variants depends on the molecular chaperone HSP70. This suggests that HSP70 recognizes the destabilized MLH1 variants and targets them for proteasomal degradation. Accordingly, we observed that several destabilized MLH1 variants associate with HSP70. Involvement of molecular chaperones in protein degradation is a well-established phenomenon (*Samant et al., 2018*; *Arndt et al., 2007*; *Kandasamy and Andréasson, 2018*). Moreover, our findings are consistent with recent developments in the field, showing that degradation signals, so-called degrons, are buried within the native structure of most globular proteins. Upon exposure when the protein structure is destabilized, the degrons are recognized by chaperones and other protein quality control components, which in turn guide the target protein for degradation (*Enam et al., 2018*; *Geffen et al., 2016*; *Kim et al., 2013*; *Maurer et al., 2016*; *Ravid and Hochstrasser, 2008*). Thus, ultimately the degradation of a protein will depend on both the structural destabilization (ΔΔG) as well as the exposed degrons, and how efficiently these are recognized by the degradation machinery. The results presented here suggest that biophysical calculations are able to predict the structural destabilization (ΔΔG), however, since the nature of protein quality control degrons is still largely undefined (*Geffen et al., 2016*; *Maurer et al., 2016*; *Rosenbaum et al., 2011*; *van der Lee et al., 2014*), in silico prediction of these is currently not possible.

In conclusion, our results support a model (*Figure 7*) where missense mutations can cause destabilization of the MLH1 protein, leading to exposure of degrons which, in turn, trigger HSP70-assisted proteasomal degradation, causing disruption of the MMR pathway and ultimately leading to an increased cumulative lifetime risk of cancer development in LS patients. Potentially, this opens up for new therapeutic approaches, including inhibiting the PQC-mediated clearance of marginally stable MLH1 variants, or small molecule stabilizers of MLH1.

## Materials and methods

**Key resources table**

| Reagent type (species) or resource | Designation | Source or reference | Identifiers | Additional information |
|---|---|---|---|---|
| Gene (*Homo sapiens*) | *MLH1* | - | UniProt identifier: P40692-1 | - |
| Cell line (*Homo sapiens*) | HCT116 | ATCC | CCL-247EMT; RRID:CVCL_0291 | - |

*Continued on next page*

*Continued*

| Reagent type (species) or resource | Designation | Source or reference | Identifiers | Additional information |
|---|---|---|---|---|
| Antibody | anti-MLH1 (rabbit polyclonal) | Santa Cruz Biotechnology | sc-11442; RRID:AB_2145332 | Dilution: 1:100 (IF) 1:1000 (WB) |
| Antibody | anti-β-actin (mouse monoclonal) | Sigma-Aldrich | A5441; RRID:AB_476744 | Dilution: 1:20000 |
| Antibody | anti-PMS2 (mouse monoclonal) | BD Biosciences | 556415; RRID:AB_396410 | Dilution: 1:2500 |
| Antibody | anti-PMS1 (rabbit polyclonal) | Invitrogen | PA5-35952; RRID:AB_2553262 | Dilution: 1:2500 |
| Antibody | anti-GFP (rat monoclonal) | ChromoTek | 3H9; RRID:AB_10773374 | Dilution: 1:2000 |
| Antibody | anti-myc (rat monoclonal) | ChromoTek | 9E1; RRID:AB_2631398 | Dilution: 1:1000 |
| Antibody | anti-HA (rat monoclonal) | Roche | 3F10; RRID:AB_2314622 | Dilution: 1:2000 |
| Antibody | anti-GAPDH (rabbit monoclonal) | Cell Signaling Technologies | 14C10; RRID:AB_10693448 | Dilution: 1:2000 |
| Antibody | anti-PMCA (mouse monoclonal) | Invitrogen | MA3-914; RRID:AB_2061566 | Dilution: 1:2000 |
| Recombinant DNA reagent | pCMV6-MYC-DDK-HSP70-1A (HSPA1A) | OriGene | RC200270 | - |
| Recombinant DNA reagent | pcDNA3-HA-HSP90 | Addgene | 22487; RRID:Addgene_22487 | - |
| Recombinant DNA reagent | pEYFP-C2-PMS2 | Prof. Lene J. Rasmussen | (*Andersen et al., 2012*) | - |
| Recombinant DNA reagent | pEGFP-C1 | Clontech | Discontinued by supplier | Available from NovoPro Labs (Cat. No. V12024) |
| Recombinant DNA reagent | pcDNA3.1-V5-His | Invitrogen | V81020 | - |
| Recombinant DNA reagent | pCMV-MLH1 and MLH1 variants | Prof. Chikashi Ishioka | (*Takahashi et al., 2007*) | - |
| Commercial assay or kit | FuGENE HD | Promega | E2311 | - |
| Chemical compound, drug | Bortezomib | LC Laboratories | B-1408 | - |
| Chemical compound, drug | YM01 | StressMarq | SIH-121 | - |
| Chemical compound, drug | Geldanamycin | Sigma-Aldrich | G3381 | - |
| Chemical compound, drug | Cycloheximide | Sigma-Aldrich | C1988 | - |
| Software, algorithm | UnScanIt gel | Silk Scientific | V6.1; RRID:SCR_017291 | - |
| Software, algorithm | FoldX | http://foldxsuite.crg.eu/ | January 2017; RRID:SCR_008522 | Details see Materials and methods |
| Software, algorithm | Gremlin | https://github.com/sokrypton/GREMLIN | V2.01 | Details see Materials and methods |
| Software, algorithm | Custom R script | - | - | - |
| Software, algorithm | SpliceAI | https://github.com/Illumina/SpliceAI | V1.2.1 | - |

## Plasmids

Plasmids for expression of wild-type and mutant MLH1 variants have been described before (*Takahashi et al., 2007*). pCMV-MYC-DDK-HSP70 and pcDNA3-HA-HSP90 were kindly provided by Dr. Kenneth Thirstrup (H. Lundbeck A/S). The pEYFP-C2-PMS2 plasmid was kindly provided by Prof. Lene J. Rasmussen (University of Copenhagen). pEGFP was purchased from Clontech. A pcDNA3-V5 vector served as a negative control.

## Cell culture

HCT116 cells (kindly provided by Prof. Mads Gyrd-Hansen (University of Oxford)) were maintained in McCoy's 5A medium (Gibco) supplemented with 10% fetal calf serum (Invitrogen), 2 mM glutamine, 5000 IU/ml penicillin and 5 mg/ml streptomycin at 37°C in a humidified atmosphere with 5% $CO_2$. Cell line authentication by STR analysis was performed by Eurofins. The cells were not contaminated by mycoplasma.

Transfections were performed using FugeneHD (Promega) as described by the manufacturer in reduced serum medium OptiMEM (Gibco). Cells were harvested no later than 72 hr after transfection at a final confluence around 90%. About 24 hr after transfection, cells were treated with serum-free growth medium containing 25 µg/mL cycloheximide (Sigma) for a duration of 4, 8 or 12 hr, 10 µM bortezomib (LC laboratories) for 8 or 16 hr, 5 µM YM01 (StressMarq) for 24 hr or 1 µM geldana-mycin (Sigma) for 24 hr. Cells were lysed in SDS sample buffer (94 mM Tris/HCl pH 6.8, 3% SDS, 19% glycerol and 0.75% β-mercaptoethanol) and protein levels were analyzed by SDS-PAGE and western blotting.

## SDS-PAGE and western blotting

Proteins were resolved by SDS-PAGE on 7 × 8 cm 12.5% acrylamide gels, and transferred to 0.2 µm nitrocellulose membranes (Advantec, Toyo Roshi Kaisha Ltd.). Blocking was performed using PBS (8 g/L NaCl, 0.2 g/L KCl, 1.44 g/L $Na_2HPO_4$, 0.24 g/L $KH_2PO_4$, pH 7.4) with 5% dry milk powder and 0.05% Tween-20. Membranes were probed with primary antibodies (see Key Resources Table) at 4°C overnight. HRP-conjugated secondary antibodies were purchased from DAKO. ECL detection reagent (GE Life Sciences) was used for development.

## Immunofluorescence and imaging

Transfected cells were seeded 24 hr prior to fixing with 4% formaldehyde in PBS in thin-bottomed 384-well plates. The fixed cells were then washed three times in PBS and permeabilized with 0.25% Triton-X-100 in PBS for 5 min at room temperature (RT). After washing with PBS, 5% bovine serum albumin (BSA, Sigma) in PBS was used for 45 min at RT for blocking. The cells were then washed with PBS and incubated with a 1:100 dilution of the anti-MLH1 antibody (Santa Cruz Biotechnology, Product no.: sc-11442) in 1% BSA in PBS for 1 hr at RT. The cells were washed with PBS and incubated with a 1:1000 dilution Alexa Fluor 568 anti-rabbit antibody (Invitrogen) in 1% BSA in PBS for 1 hr at RT. After additional washing with PBS, the DNA was stained with Höchst 33342 (Sigma) for 10 min. Microscopy was performed using an InCell2200 microscope (GE Healthcare). The filters were Höchst (ex 390 nm, em 432 nm) and TexasRed (ex 575 nm, em 620 nm). The InCell Developer Toolbox (GE Healthcare) was used for image analysis. To determine the abundance of the MLH1 variants, the total intensity of the red channel in each cell was measured after excluding the non-transfected cells.

## Co-immunoprecipitation

Transfected cells were lysed in buffer A (50 mM Tris/HCl pH 7.5, 150 mM NaCl, 1 mM EDTA and 0.5% NP-40 supplemented with Complete Mini EDTA-free Protease inhibitor cocktail tablets (Roche)) and left to incubate for 20 min on ice. The lysates were cleared by centrifugation (13000 g, 30 min) and the proteins were captured with GFP-trap (Chromotek), Myc-trap (Chromotek) or HA-agarose beads (Sigma) by tumbling end-over-end overnight at 4°C. The beads were washed three times by centrifugation (1000 g, 10 s) in buffer A. Finally, the beads were resuspended in SDS sample buffer (94 mM Tris/HCl pH 6.8, 3% SDS, 19% glycerol and 0.75% β-mercaptoethanol) and analyzed by SDS-PAGE and western blotting.

## Solubility assays

For analyses of wild-type MLH1, HCT116 cells were transfected as described above. Cells were harvested in buffer B (50 mM Tris/HCl pH 7.5, 150 mM NaCl, 1 mM EDTA) supplemented with complete protease inhibitor cocktail tablets (Roche), and lysed by sonication (3 × 10 s). Then the lysate was distributed into different tubes that were incubated at 30 min. at the indicated temperatures. The soluble and insoluble fractions were separated by centrifugation (15000 g, 4℃, 30 min), after which the supernatant was removed and the pellet washed once in buffer B. Subsequently, SDS sample buffer was added, and the final sample volume was kept identical between the pellet and the supernatant. Finally, fractions were analyzed by SDS-PAGE and western blotting as described.

For comparison of the variants, HCT116 cells were transfected and treated with bortezomib as described above. Cells were harvested in buffer B supplemented with complete protease inhibitor cocktail tablets (Roche), and lysed by sonication (3 × 10 s). The soluble and insoluble fractions were then immediately separated by centrifugation and analyzed SDS-PAGE and western blotting as described above.

## Stability calculations

The changes in folding stability ($\Delta\Delta G$) were calculated using FoldX (*Guerois et al., 2002*) based on PDB IDs 4P7A (*Wu et al., 2015*) for the N-terminal domain of MLH1, and 3RBN for the C-terminal domain. The $\Delta\Delta G$s were calculated from each structure individually by first applying the RepairPDB function to fix minor issues in the original coordinates, and then the BuildModel function to generate each individual amino acid variant. Each calculation was repeated five times, and the average difference in stability between wild type and variant is reported, such that values < 0 kcal/mol indicate stabilized variants, and values > 0 kcal/mol indicate destabilized variants with respect to the wild type MLH1 protein. Values > 15 kcal/mol likely indicate clashes in the model FoldX generated. While this does indicate that major destabilization is likely, the actual values are less meaningful for these clashing variants, and we thus truncated them to 15 kcal/mol in our figures.

## Evolutionary sequence energy calculations

To assess the likelihood of finding any given variant in the protein family, we created a multiple sequence alignment of human MLH1 using HHblits (*Zimmermann et al., 2018*) and then calculated a sequence log-likelihood score combining site-conservation and pairwise co-variation using Gremlin (*Balakrishnan et al., 2011*). Scores were normalized by their rank to a range of (0,1), with low scores indicating tolerated sequences and high scores indicating variants that are rare or unobserved across the multiple sequence alignment. Positions at which the number of distinct homologous sequences was too small to extract meaningful evolutionary sequence energies are set to NA (*Supplementary file 2*). Other sequence-based predictions of functional variant consequences were retrieved from the webservers of PROVEAN (*Choi and Chan, 2015*) and PolyPhen2 (*Adzhubei et al., 2010*), and extracted from the pre-calculated scores for REVEL (*Ioannidis et al., 2016*).

## Dominant mutator effect (DME)

We grouped the functional classification observations from *Takahashi et al. (2007)* into four categories by summarizing the number of assays each variant showed functional behavior in. Thus, variants in group 0 were non-functional in all three assays, those in group three were functional in all three assays, and the rest were functional in some, but not in other assays.

## Acknowledgements

The authors thank Prof. Lene J Rasmussen, Prof. Mads Gyrd-Hansen, and Dr. Kenneth Thirstrup for sharing cells and reagents, and Dr. Elin J Pietras, Dr. Cornelia Steinhauer, Cecilie Søltoft, Anne-Marie Lauridsen, and the Core Facility for Integrated Microscopy for excellent technical assistance.

## Additional information

### Funding

| Funder | Grant reference number | Author |
|--------|------------------------|--------|
| Novo Nordisk Foundation | PRISM Challenge Program | Amelie Stein<br>Kresten Lindorff-Larsen<br>Rasmus Hartmann-Petersen |
| Novo Nordisk Foundation | Young Investigator Award, NNF15OC0016662 | Eva R Hoffmann |
| The A.P. Møller Foundation | Project Grant | Rasmus Hartmann-Petersen |
| Danish Cancer Society | Project Grant | Rasmus Hartmann-Petersen |
| Danish Council for Independent Research | Project Grant | Rasmus Hartmann-Petersen |
| Lundbeck Foundation | Project Grant | Kresten Lindorff-Larsen<br>Rasmus Hartmann-Petersen |
| Lundbeck Foundation | Fellowship Grant | Amelie Stein |
| Novo Nordisk Foundation | Hallas-Møller Program | Kresten Lindorff-Larsen |

The funders had no role in study design, data collection and interpretation, or the decision to submit the work for publication.

### Author contributions

Amanda B Abildgaard, Formal analysis, Investigation, Visualization, Writing—original draft; Amelie Stein, Formal analysis, Investigation, Visualization, Writing—review and editing; Sofie V Nielsen, Formal analysis, Methodology, Writing—review and editing; Katrine Schultz-Knudsen, Formal analysis, Investigation, Writing—review and editing; Elena Papaleo, Conceptualization, Investigation; Amruta Shrikhande, Supervision, Investigation; Eva R Hoffmann, Conceptualization, Supervision; Inge Bernstein, Anne-Marie Gerdes, Conceptualization, Writing—review and editing; Masanobu Takahashi, Chikashi Ishioka, Resources, Writing—review and editing; Kresten Lindorff-Larsen, Rasmus Hartmann-Petersen, Conceptualization, Formal analysis, Supervision, Writing—review and editing

### Author ORCIDs

Amelie Stein https://orcid.org/0000-0002-5862-1681
Elena Papaleo https://orcid.org/0000-0002-7376-5894
Kresten Lindorff-Larsen https://orcid.org/0000-0002-4750-6039
Rasmus Hartmann-Petersen https://orcid.org/0000-0002-4155-7791

### Decision letter and Author response

Decision letter https://doi.org/10.7554/eLife.49138.023
Author response https://doi.org/10.7554/eLife.49138.024

## Additional files

### Supplementary files

• Supplementary file 1. Full matrix for FoldX stability predictions.
DOI: https://doi.org/10.7554/eLife.49138.019

• Supplementary file 2. Full matrix for evolutionary sequence energies.
DOI: https://doi.org/10.7554/eLife.49138.020

• Transparent reporting form DOI: https://doi.org/10.7554/eLife.49138.021

### Data availability

All data generated or analysed during this study are included in the manuscript and supporting files.

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
