## [Decision Letter]

**Acceptance summary:**

This contribution presents a comprehensive computational-experimental analysis of mutations in the *MLH1* gene, which is involved in DNA mismatch repair, and their relationship to protein stability, association with chaperones and cellular expression levels. The manuscript reports a large array of computational and experimental methods, including computational mutation scanning, structural analysis, evolutionary conservation, clinical relevance, cellular localization and expression levels, and chaperone association. Although the association between protein stability and disease has a very long history in molecular biology, the ability to separate potentially disease-causing mutations from benign ones based on energetics is important and the large-scale analysis reported here may in the future enable improved predictive and even diagnostic capabilities.

**Decision letter after peer review:**

Thank you for submitting your article "Computational and cellular studies reveal structural destabilization and degradation of MLH1 variants in Lynch syndrome" for consideration by *eLife*. Your article has been reviewed by three peer reviewers, including Sarel Jacob Fleishman as the Reviewing Editor and Reviewer #1, and the evaluation has been overseen by John Kuriyan as the Senior Editor.

The reviewers have discussed the reviews with one another, and the Reviewing Editor has drafted this decision to help you prepare a revised submission.

Summary:

Lynch syndrome is an inherited disorder, associated with a genetic predisposition to certain cancers. A major cause for the development of Lynch disease is the loss of function of the MLH1 protein, which is an essential component of the cellular defective mismatch repair (MMR) complex. In this manuscript, Abildgaard and co-workers used various computational, cellular and biochemical tools to test the link between structural destabilization of disease-linked variants of MLH1, their steady-state levels, and pathogenicity. The authors show that levels of MLH1 variants with lower thermodynamic folding stability (ΔΔG) are reduced, compared to the wild-type protein. This is largely due to their HSP70 chaperone-assisted degradation by the proteasome. Comparison of the prediction of structural changes, due to thermodynamic instability of missense MLH1 variants, to sequence-based tools currently employed in the clinic, indicate that the former is slightly better at distinguishing pathogenic and non-pathogenic variants. Although the association between protein stability and disease has a very long history in molecular biology, the ability to separate potentially disease-causing mutations from benign ones based on energetics is important, and the large-scale analysis reported here may in the future enable improved predictive and even diagnostic capabilities.

This is a well written, coherent study that highlights the potential of missense mutations to trigger protein destabilization, thereby to reduce MLH1 activity, resulting in a disease phenotype. Notably, a similar approach was previously taken by the same research group to explore the role of missense mutations in the destabilization of phenylalanine hydroxylase, leading to phenylketonuria. In the current manuscript, the authors delve deeper into the mechanism of MLH1 pathogenicity, further demonstrating the effects of unstable variants on the stability of PMS1 and PMS2, which co-operate with MLH1 in the MMR complex. The authors also identified HSP70 (but not HSP90) chaperones as key players in the degradation pathway. Altogether, these studies provide new and exciting insights about the mechanisms of certain pathogenies, which are based on structural instability, leading to reduced protein levels.

Essential revisions:

1) The authors linked structural instability, chaperone requirements and proteasomal degradation to reduced levels of pathogenic MLH1 variants. It is unclear, however, how local protein instability leads to proteasomal degradation. One obvious possibility is that major thermodynamic instability leads to protein misfolding. None of the tested variants, however, form notable inclusion bodies in the cell, which is the hallmark of protein misfolding diseases. Hence, in order to show a link between ∆∆G values, protein misfolding and proteasomal degradation the authors can test the fate MLH1 mutants when degradation is inhibited. This can be done by testing the formation of inclusion bodies (via fluorescence microscopy or protein solubility assays) upon proteasome/lysosome inhibition or upon inhibition of HSP70. The accumulation of protein aggregates will be indicative of protein misfolding, thus explaining the need for HSP70 chaperones. Also, in Figure 5E, F, changes in MLH1 steady-state levels upon chaperone inhibition should be quantified and compared to control no treatment as done for Figure 5A, C. Otherwise, it is difficult to discern the differences.

2) FoldX is used to predict ∆∆G values of individual mutations, but Figure 6D suggests that it provides only a minor benefit above standard evolutionary conservation calculations (AUC of 87 and 84%, respectively). The main advantage of evolutionary conservation analysis is that it is independent of structures which are often not available. The authors should comment on whether FoldX represents a sufficiently substantial improvement over evolutionary analysis to warrant the added complication of relying on a structure. Moreover, Figure 6E suggests that in fact the two analyses could be combined to provide a better analysis of harmful/benign mutations but there are no statistics provided for such a combination. Could the authors use logistic regression or similar to suggest cutoffs that use both terms (with appropriate statistical controls, such as cross-validation)? If this analysis is superior to FoldX alone, the message of the paper may need to change from stability to stability+conservation analysis.

3) Somewhat conflicting themes run through the paper. On the one hand, the authors note in several places that misfolding is likely to be an important consequence of destabilizing mutations; yet, in many places, the reference state for the mutations is the unfolded state (Figure 1C, Figure 7, and main text). Additionally, the authors note that above a predicted value of ∆∆G>3kcal/mol, mutations are harmful, finding this low value to be surprising (subsection “Thermodynamic stability calculations predict severely reduced MLH1 steady-state levels”, third paragraph). Have they considered whether the MLH1 protein is marginally stable, which would reconcile these observations. If the protein is not marginally stable, then its intrinsic stability would be predicted to buffer the effects of many of the single-point mutations. The best way to address this question is to show (if this has not been done before) a standard denaturation assay of the protein (temperature melt, for instance). It may also be beneficial to clearly state and illustrate (in the figures showing the reference states) that the misfolded states are likely to be the dominant reference for the mutants and that the predictive value of ∆∆G calculations is likely to be relevant to marginally stable proteins.

4) Additionally, the link that the authors claim between protein stability and cellular expression levels is not absolutely solid since there are numerous layers of control of protein expression in mammalian cells. As noted above, this link is likely to apply more to marginally stable proteins. For the benefits of the readers, it would be good if a more elaborate discussion of this relationship is presented already in the Introduction.

5) The authors should make it clearer that the approach and findings in this manuscript are quite similar to their previous paper on MSH2 (Nielsen et al., 2017) as this is not immediately apparent from the manuscript.

---

## [Author Response]

Essential revisions:1) The authors linked structural instability, chaperone requirements and proteasomal degradation to reduced levels of pathogenic MLH1 variants. It is unclear, however, how local protein instability leads to proteasomal degradation. One obvious possibility is that major thermodynamic instability leads to protein misfolding. None of the tested variants, however, form notable inclusion bodies in the cell, which is the hallmark of protein misfolding diseases. Hence, in order to show a link between ∆∆G values, protein misfolding and proteasomal degradation the authors can test the fate MLH1 mutants when degradation is inhibited. This can be done by testing the formation of inclusion bodies (via fluorescence microscopy or protein solubility assays) upon proteasome/lysosome inhibition or upon inhibition of HSP70. The accumulation of protein aggregates will be indicative of protein misfolding, thus explaining the need for HSP70 chaperones. Also, in Figure 5E, F, changes in MLH1 steady-state levels upon chaperone inhibition should be quantified and compared to control no treatment as done for Figure 5A, C. Otherwise, it is difficult to discern the differences.

We tested the solubility of selected MLH1 variants with and without proteasome inhibitor by separating crude cell lysates into soluble and insoluble fractions by centrifugation. The results show that the destabilized variants are more insoluble than the stable variants, and bortezomib treatment appears to mainly result in accumulation of insoluble MLH1. In the revised manuscript, we discuss these results in the last paragraph of the subsection “Proteasomal degradation causes reduced steady-state levels of destabilized MLH1 variants” and include them as Figure 3—figure supplement 1. We find these additional data supports the notion that the degraded MLH1 variants are structurally destabilized, and that under normal conditions these are cleared by the protein quality control system (rather than forming substantial aggregates).

In addition, we added quantifications of the blots in Figure 5 (subsection “HSP70 is required for degradation of some destabilized MLH1 variants” and Figure 5 legend).

2) FoldX is used to predict ∆∆G values of individual mutations, but Figure 6D suggests that it provides only a minor benefit above standard evolutionary conservation calculations (AUC of 87 and 84%, respectively). The main advantage of evolutionary conservation analysis is that it is independent of structures which are often not available. The authors should comment on whether FoldX represents a sufficiently substantial improvement over evolutionary analysis to warrant the added complication of relying on a structure. Moreover, Figure 6E suggests that in fact the two analyses could be combined to provide a better analysis of harmful/benign mutations but there are no statistics provided for such a combination. Could the authors use logistic regression or similar to suggest cutoffs that use both terms (with appropriate statistical controls, such as cross-validation)? If this analysis is superior to FoldX alone, the message of the paper may need to change from stability to stability+conservation analysis.

The reviewer correctly asserts that the stability calculations by FoldX provide a similar prediction accuracy (AUC~0.82) to the sequence analysis (AUC~0.88) (which in this case takes both site conservation and context into account). Thus, from a purely predictive perspective, these two methods are comparable. We argue, however, that the stability calculations – because they focus on a single molecular effect (change in protein stability) – are useful to help support our observation of Lynch syndrome as a “misfolding/unfolding” disease. Thus, the strength lies in the mechanistic argument and not necessarily in the purely predictive value. Indeed, the two-dimensional analysis in Figure 6D shows that most non-functional variants (DME=0) lie in the upper-right quadrant suggesting that they are unstable and that this why the evolutionary analysis also find these to be likely pathogenic. This is the reason why we generally kept the two analyses (stability and conservation) separate – so that we can examine whether those variants that are predicted to be poor from a conservation analysis are so because of the lack of stability. In the revised manuscript we make this point clearer (Discussion). As suggested by the reviewers, we have also developed a combined predictor using logistic regression on FoldX and evolutionary analysis. As Figure 6F shows, performance of the combined model is above either of the individual predictors, though not by much (AUC~0.90). As we have also previously pointed out, and as the 2D plots illustrate (Figure 6D, E, G), most pathogenic and non-functional variants are both destabilized and rare in the alignment. We hypothesize that they are selected against because those variants would destabilize the protein. Nevertheless, a few pathogenic/non-functional variants are only identified by one of the predictors, and the linear regression model includes several of these from both predictors, thus gaining in overall performance. We tested performance of the regression model in a leave-one-out jackknife procedure, which correctly classified 85% of the respective data points when they had been left out of the training set (see subsection “Structural stability calculations for predicting pathogenic mutations”).

3) Somewhat conflicting themes run through the paper. On the one hand, the authors note in several places that misfolding is likely to be an important consequence of destabilizing mutations; yet, in many places, the reference state for the mutations is the unfolded state (Figure 1C, Figure 7, and main text). Additionally, the authors note that above a predicted value of ∆∆G>3kcal/mol, mutations are harmful, finding this low value to be surprising (subsection “Thermodynamic stability calculations predict severely reduced MLH1 steady-state levels”, third paragraph). Have they considered whether the MLH1 protein is marginally stable, which would reconcile these observations. If the protein is not marginally stable, then its intrinsic stability would be predicted to buffer the effects of many of the single-point mutations. The best way to address this question is to show (if this has not been done before) a standard denaturation assay of the protein (temperature melt, for instance). It may also be beneficial to clearly state and illustrate (in the figures showing the reference states) that the misfolded states are likely to be the dominant reference for the mutants and that the predictive value of ∆∆G calculations is likely to be relevant to marginally stable proteins.

We agree with the reviewer that this was somewhat confusing, and have attempted to be clearer about this issue in the revised manuscript. Due to the size of MLH1 (84.6 kDa) and its binding partner PMS2 (95.8 kDa) a careful in vitro analysis to obtain an absolute measure of MLH1’s thermodynamic stability would be rather difficult to perform. In the revised manuscript we therefore include new data where we have followed the solubility of wild-type MLH1 over a range of temperatures (Figure 2—figure supplement 1). These data clearly show, as the reviewers suspected, that MLH1 is somewhat less stable than the most abundant cellular proteins (Ponceau S stained) and GAPDH. Hence, it is possible that MLH1 is marginally stable. However, under normal conditions wild-type MLH1 is clearly not so unstable that it is rapidly degraded. Our analyses indicate that MLH1 has a half-life > 12 hours, which is in line with previous studies showing a t½ of about 40 hours in B-cells (Mathieson et al. 2018 Nat. Commun. 9:689). In yeast, wild-type MLH1 is also a stable protein, with a cellular stability similar to that of proteasome subunits, histones H3/H4 and certain ribosomal proteins (Christiano et al. 2014 Cell Rep. 9:1959-1965). We agree that stable proteins could buffer the effect of mutations at least in terms of in vitro stability, but a number of recent studies have shown that in many cases a sizable fraction of single amino acid substitutions may render an otherwise perfectly stable protein unstable in cells (e.g. Matreyek et al., 2018; Suiter et al., 2019, bioRxiv https://doi.org/10.1101/740837). Accordingly, it appears that the PQC system is tightly tuned to detect even minor or transient structural defects, which is supported by observations that in some cases the destabilized variants are even functional when the PQC system is blocked (e.g. Arlow et al., 2013). We have included a discussion of this point in the manuscript (Discussion). Since Figure 1B, C describe the ΔΔG predictions that refer to the unfolded state, we have not made any changes to Figure 1. However, in Figure 7, which summarizes the fate for MLH1 variants in the cell we have change “unfolded” to “non-native”, since we fully agree that none of the MLH1 variants are likely to fully unfold within a living cell (also discussed in the Discussion).

Regarding the last point, it remains unclear for which types of proteins these kinds of calculations will be useful to predict pathogenicity and cellular abundance, and to which extent the calculations of effects on global stability are correlated with more subtle conformational changes. We note, however, that we have made similar observations in other proteins (MSH2, phenylalanine hydroxylase, PTEN) that differ in size and likely intrinsic stability. We expect that while our stability calculations use the unfolded state as reference, these values will be correlated with effects on local stability effects that may underlie cellular degradation, though future studies that combine more detailed biophysical experiments are likely needed to answer this more quantitatively. In the revised manuscript we discuss this issue, and also draw parallels to related observations other proteins.

In addition, we have amended the text and Figure 7 so they more clearly reflect that the misfolded state is likely the dominant state for the destabilized MLH1 variants.

4) Additionally, the link that the authors claim between protein stability and cellular expression levels is not absolutely solid since there are numerous layers of control of protein expression in mammalian cells. As noted above, this link is likely to apply more to marginally stable proteins. For the benefits of the readers, it would be good if a more elaborate discussion of this relationship is presented already in the Introduction.

We agree that the situation in a human being will be substantially more complicated than in the cellular system we studied, and that the biophysical calculations only capture some of the effects seen in cells. Related to the specific point the reviewer makes, several of the effects from potential feedback mechanisms have purposely been removed to focus on a single effect. Specifically, we study protein levels from MLH1 expressed from a constitutive promotor, and under conditions where MLH1 function is not essential. Thus, we do not expect MLH1-specific feedback mechanisms to play a substantial role. Nevertheless, we agree that MLH1 misfolding is likely to lead to a cellular response including potentially induction of chaperones and other PQC components. Thus, the abundance levels and degradation rates we measure should be seen in this background, and thus report on the “effective stability” in the cell, which – as the reviewer suggests – may be different from the predicted thermodynamic effect. We now elaborate on this in the revised Discussion.

5) The authors should make it clearer that the approach and findings in this manuscript are quite similar to their previous paper on MSH2 (Nielsen et al., 2017) as this is not immediately apparent from the manuscript.

We now stress this point in the manuscript (Introduction and Discussion).